

# Impact of a Strong Biomass Burning Event on the Radiative Forcing in the Arctic

Justyna Lisok[1], Anna Rozwadowska[2], Jesper G. Pedersen[1], Krzysztof M. Markowicz[1], Christoph Ritter[3], Jacek W. Kaminski[4], Joanna Struzewska[5], Mauro Mazzola[6], Roberto Udisti[6,7], Silvia Becagli[7], and Izabela Górecka[8]

[1]Institute of Geophysics, Faculty of Physics, University of Warsaw, Poland
[2]Institute of Oceanology, Polish Academy of Sciences, Sopot, Poland
[3]Alfred Wegener Institute for Polar and Marine Research, Potsdam, Germany
[4]Department of Atmospheric Physics, Institute of Geophysics Polish Academy of Science, Warsaw, Poland
[5]Faculty of Building Services Hydro and Environmental Engineering, Warsaw University of Technology, Warsaw, Poland
[6]National Research Council, Institute of Atmospheric Sciences and Climate, Bologna, Italy
[7]Department of Chemistry, University of Florence, Florence, Italy
[8]Geoterra, Gdańsk, Poland

*Correspondence to:* Justyna Lisok (jlisok@igf.fuw.edu.pl)

**Abstract.** The aim of the presented study was to investigate the impact on the radiation budget of biomass burning smoke plume transported from Alaska to high Arctic region (Ny-Alesund, Svalbard) in early July 2015. This high aerosol load event is considered exceptional in the last 25 years with mean aerosol optical depth increased by the factor of 10 in comparison to the average summer background values. We utilised in-situ data with hygroscopic growth equations as well as remote sensing

measurements as inputs to radiative transfer models with an objective to estimate biases associated with (i) hygroscopicity, (ii) variability of $\omega$ profiles and (iii) plane-parallel closure of the modelled atmosphere. A chemical weather model with satellite-derived biomass burning emissions was used to interpret the transport and transformations pathways.

Provided MODTRAN simulations resulted in the mean aerosol direct radiative forcing on the level of -78.9 $\mathrm{Wm^{-2}}$ and -47.0 $\mathrm{Wm^{-2}}$ at the surface and the top of the atmosphere respectively for the mean value of aerosol optical depth equal to 0.64 at

550 nm. It corresponded to the average clear-sky direct radiative forcing of -43.3 $\mathrm{Wm^{-2}}$ estimated by radiometers and model simulations. Furthermore, model-derived aerosol direct radiative forcing efficiency reached on average -126 $\mathrm{Wm^{-2}}/\tau_{550}$ and -71 $\mathrm{Wm^{-2}}/\tau_{550}$ at the surface and at the top of the atmosphere. Estimated heating rate up to 1.8 $\mathrm{Kday^{-1}}$ inside the BB plume implied vertical mixing with turbulent kinetic energy of 0.3 $\mathrm{m^2s^{-2}}$. Ultimately, uncertainty connected with the plane-parallel atmosphere approximation altered results by about 2 $\mathrm{Wm^{-2}}$.

*Copyright statement.* TEXT



# 1 Introduction

Wildfires are considered a significant source of carbon in the atmosphere. It is estimated that up to 2.0 Pg of carbon aerosol is released to the atmosphere each year (Van der Werf et al., 2010). In the past 100 years, an intensification of fires in mid-latitudes is observed affecting appreciably radiative and optical properties of the atmosphere (Mtetwa and McCormick, 2003). The

emitted particles from biomass burning (BB) sources mainly consist of organic and black carbon (Smithson, 2002), which are in 90% built of the fine mode regarding aerosol size distribution (Dubovik et al., 2002). The impact of the plume on the atmospheric instability conditions and its rather small particle radius might result in a rapid transport on an intercontinental scale within just several days (Nikonovas et al., 2015). Thus, it is likely that the biomass burning aerosol affects appreciably the optical and radiative properties of the atmosphere in the substantial part of the globe. The influence of BB aerosol is manifested

by heating the air layer where transport takes place. Regarding the columnar properties however, it implies weak cooling effects at the top of the atmosphere (TOA) due to predominant scattering properties of the plume (Hansen et al., 2004). The magnitude of its impact is nevertheless strongly dependent on the chemical composition, which results from the adversative radiative response of the atmosphere exposed to black and organic carbon being negative for the latter (Myhre et al., 2013).

  A number of papers analysed the associated annual mean of instantaneous clear-sky aerosol direct radiative forcing ($RF$)

at TOA. Myhre et al. (2013) presented the results from AeroComII 28 models indicating global mean BB $RF_{toa}$ on the level of -0.01 ± 0.08 Wm$^{-2}$. The very similar value was presented by Fifth Assessment Report, IPCC in Pachauri et al. (2014) being equal to 0.0 ± 0.2 Wm$^{-2}$. The effect of BB aerosol from the regional point of view is claimed to have stronger temporal variations indicating the change of the regional climate patterns (Wang et al., 2006). It might be especially important over the bright surfaces regarding changes in the surface and cloud albedo (Screen and Simmonds, 2010), which in particular may

indicate a positive $RF_{toa}$.

  The existing discrepancies in the model's results (e.g., Myhre et al., 2013) regarding the impact of biomass burning aerosol on the radiative budget come from various difficulties in estimation of an input data (Carslaw et al., 2010). According to Koch et al. (2009) and Bond et al. (2013) the most significant uncertainty is recognised in parametrisation of mixing effects and ageing processes of the plume that influence optical and microphysical properties of the aerosol. Therefore, the most

meaningful parameters in the light of radiative transfer calculations are single scattering albedo ($\omega$) indicating relation between scattering and absorbing properties of the plume as well as asymmetry parameter ($g$) as a simple representation of a phase function. When BB aerosol becomes increasingly older, it suffers great changes, usually by increasing the values of the cited parameters (Ortiz-Amezcua et al., 2017; Janicka et al., 2017), thus simultaneously losing gradually the ability to absorb the light and expanding the particle size (Nikonovas et al., 2015). In order to parametrize the single-scattering properties of the plume,

one needs to use measurement data of $\omega$ and aerosol size distribution. The transformation of $\omega$ and $g$ might be investigated by inversion schemes utilising sun-photometer data installed at AERONET network, however, the uncertainty of the columnar $\omega$ retrieval becomes high considering low levels of aerosol optical depth ($\tau$) values (Dubovik et al., 2000). This is the reason why AERONET level 2 data validation is performed only for $\tau_{440}$ more than 0.5 and solar zenith angle above 50$^{\text{o}}$ (Dubovik et al., 2002). This, in turn, leads to a significant reduction of values calculated for Arctic region (Markowicz et al., 2017a).





$\omega$ and $g$ may also be calculated utilising in-situ measurements, however, one needs to account for the parameter's translation from dry to ambient conditions. The impact of water uptake by aerosol is significant regarding soluble particles when exposed to the relative humidity RH more than 40% resulting in the enhancement of particle scattering cross section (Orr et al., 1958). Some studies apply empirical formulas of an enhancement factor $f(RH)$ to retrieve the hygroscopic properties of

particles (Kotchenruther and Hobbs, 1998). Considering fresh plume of biomass burning aerosol, $f(RH)$ was found to be 1.1 for RH values not exceeding 80%, while around 1.35 for the aged smoke. This transition is in agreement with secondary production processes resulting in sulphate formation and progressive oxidation of organic compounds with OH and COOH groups increasing the hygroscopic properties (Reid et al., 2005). However, the values of growth factor might vary significantly due to the particle chemical composition related to emission source (Gras et al., 1999; Magi et al., 2003; Kreidenweis et al.,

2001) and additionally due to particle size (Carrico et al., 2010).

In July 2015 the intense tundra and boreal forest fires in the northern part of North America together with meteorological conditions indicated a direct transport of biomass-burning plume over Arctic region, altering both the optical and microphysical properties of aerosols. Thus, $\tau$ conditions characteristic to summer conditions were changed with a magnitude of 10 determining it, the strongest event in 25 years (Markowicz et al., 2016a). Markowicz et al. (2016a) reports the development and further

intensification of tundra fires in Alaska introduced by series of frequent lightning strikes occurring from the mid-June to late July 2015. The transport of BB plume was visible between $4^{th}$ and $6^{th}$ July from the central part of Alaska via the North Pole to the Spitsbergen. Starting with the afternoon $9^{th}$ till around noon $11^{th}$ July the BB plume was visible in Ny-Alesund as indicated by in-situ and remote sensing instruments. The mean $\tau_{550}$ value for the event was estimated at the level of 0.64, while Ångström exponent (AE) was around 1.5 indicating an existence of mostly small particles (Markowicz et al., 2016a). This

hypothesis is confirmed by the size distribution measured at the ground level, which shows that particles during the BB event are mainly distributed in the accumulation mode. Also, a significant increase in the precipitable water (PW) was measured with the mean value of 2.2 cm for the entire event. It was especially notable due to the hygroscopic properties of aerosols. The extensive optical properties of aerosols measured by in-situ instruments significantly exceeded typical annual mean values reported in Schmeisser et al. (2017) for a couple of stations in the Arctic. The absorption coefficient ($\sigma_{abs}$) reached 4.0 Mm$^{-1}$,

while extinction coefficient ($\sigma_{ext}$) 65.0 Mm$^{-1}$ on average.

This study presents the impact of BB aerosol advected from the Alaskan wildfires in July 2015 on the clear-sky radiative budget and atmospheric dynamics. We included implementation of new methods to retrieve the profile of $\omega$ at ambient conditions utilising in-situ measurements and lidar profiles (section 3.1) and compared obtained values of RF with the more robust model (section 3.3). Section 3.4 shows an example of RF distribution at the surface in the vicinity of Ny-Alesund.

The last part presents the influence of unstably stratified biomass burning air masses on the turbulence development, which is shown in chapter 3.5. The research aims to estimate the biases connected with (i) hygroscopicity, (ii) variability of $\omega$ profiles and (iii) plane-parallel closure of the modelled atmosphere. Additionally, we confirmed the source region of the BB plume. A chemical weather model with satellite-derived biomass burning emissions was used to interpret the transport and transformations pathways.



## 2 Methodology

This chapter consists of a few subsections dedicated to a brief description of all data and models used in this research. In paragraph 2.1 we will focus on describing all models used to track the transport of smoke as well as to calculate the impact of the BB plume. Regarding the latter, having in mind model's possibilities and the computational costs due to radiative

transfer equation approximations, we decided to use MODTRAN 5.2.1 only for a small-scale, 3D Monte-Carlo for semi-small and Fu-Liou for large-scale calculations. All results from Fu-Liou calculations are presented in Markowicz et al. (2017b), however here we wanted to compare them with the results from our retrieval in MODTRAN. A number of different inputs for models were prepared. Regarding small and semi-small scales, we used data provided by in-situ and remote sensing instruments (section 2.2) to retrieve vertical profiles of optical properties (section 2.3.1). For the latter, we retrieved maps

of optical properties from MODIS products and NAAPS model (for more details see Markowicz et al., 2017b).

### 2.1 Modelling tools

The MODerate-resolution atmospheric radiance and TRANsmittance model (MODTRAN) v. 5.2.1 (Berk et al., 1998) is the radiative transfer model. In this study, the model uses 8 streams discrete ordinate radiative transfer (DISORT) code to obtain the radiation fluxes with a spectral resolution of $15\,\mathrm{cm^{-1}}$ (Stamnes et al., 1988). Simulations were run with defined 17 absorption

coefficients for each band in correlated-k method also handling with the multiple scattering (Bernstein et al., 1996) as well as Henyey-Greenstein scattering phase function approximation (Henyey and Greenstein, 1941). Calculations are performed for the user-defined vertical profiles of thermodynamical variables and aerosol optical properties as well as the optical properties of trace gases provided by the HITRAN 2000 database (Rothman et al., 1998). MODTRAN was run with the time resolution of 20 min from 9 to 11 July 2015 for the domain set to Ny-Alesund coordinates. Simulations were set up to calculate cases with and

without aerosol load (polluted and reference). Additionally, 2 sets of simulations were run with a different spectral dependency of surface albedo, which included the values obtained from the radiometers (real albedo case) and Fresnel calculations of water surface reflectance (ocean albedo case). Results were used to intercompare them with measurements (first) and with Fu-Liou simulations (the latter).

The Fu-Liou v. 200503 (Fu and Liou, 1992, 1993) model uses the $\delta$ 2/4 stream solver applied for 6 shortwave and 12

longwave spectral bands. The optical properties of the atmosphere are calculated by the correlated-k distribution 2.4 method defined for each spectral band (Fu and Liou, 1992). The optical properties of aerosols, as well as thermodynamical properties of the atmosphere, were based on the results provided by the NAAPS re-analysis (Lynch et al., 2016). Fu-Liou was used to determine the spatial distribution of RF for the period 5 - 15 July 2015 in the area to the north of $55^{\circ}$N, where the transport of BB aerosol was observed. The results of these simulations are presented in Markowicz et al. (2017b). In this paper, we compare

the results of our retrieval from MODTRAN simulations and robust Fu-Liou code over the area close to Ny-Alesund.

The Navy Aerosol Analysis and Prediction System (NAAPS) is the semi-lagrangian aerosol transport model run on a $1^{\circ}$ x $1^{\circ}$ grid and 6 h temporal resolution and with 28 vertical layers (Hogan and Rosmond, 1991). NAAPS model utilises 5 basic processes including emission from the source, mixing and diffusion within the PBL, dispersion and advection due to




the wind as well as wet and dry deposition (Lynch et al., 2016). Recently, a new version of NAAPS model was implemented, which assimilates quality controlled $\tau$ values retrieved from MODIS and MISR products (Hyer et al., 2011; Shi et al., 2014). Furthermore, thermodynamics fields are taken from the Operational Global Analysis and Prediction System (NOGAPS; Hogan and Rosmond, 1991). The results were used as inputs to Fu-Liou simulations.

3D effects of the RF were calculated using 3D forward Monte Carlo code (Marshak et al., 1995), which utilises maximum cross-section method to compute photon paths in the three-dimensional model of the atmosphere (Marchuk et al., 2013). A number of modifications were made to the original setup of the code, including such phenomena like absorption of photons by atmospheric gases and reflection and absorption at the undulated Earth's surface (Rozwadowska and Górecka, 2012, 2017). The model domain covers the area of 51 km (W-E axis) x 68 km (S-N axis) and consists of cells/columns of 200 x 200 m.

The main domain is surrounded by 20 km wide belt in order to reduce the impact of cyclic boundaries on the results in the Monte Carlo modelling. The computations were performed for the whole 91 x 108 km domain, but only the results from the main domain were analysed. The Earth's surface was represented by the Digital Elevation Model (DEM) and the technique proposed by Ricchiazzi and Gautier (1998).

Implicit large-eddy simulations (ILESs) were performed using the 3D nonhydrostatic anelastic Eulerian-semi-Lagrangian

(EULAG) model (Prusa et al., 2008) to estimate the dynamical response of the atmosphere induced by the BB plume. The EULAG model is set up to solve for the three velocity components u, v, and w in the x-, y-, and z-directions (i.e. W-E, S-N, and vertical directions), and for the potential temperature ($\theta$). The governing equations are solved in an Eulerian framework without explicit subgrid-scale (SGS) terms included. The nonoscillatory forward-in-time integration is performed with the Multidimensional Positive definite Advection Transport Algorithm (MPDATA; Smolarkiewicz, 2006), and we rely on the

ability of the MPDATA to implicitly account for the effect of unresolved turbulence on the resolved flow through the truncation terms associated with the algorithm. For more details on ILES, see e.g. Grinstein et al. (2007). The horizontal grid spacing was set to 200 m and the vertical grid spacing to 50 m. The size of the computational domain was set to 19 km in the horizontal directions and 20 km in the vertical direction. The uppermost 5 km is a sponge layer included to prevent reflection of gravity waves at the top of the domain. The upper boundary of the domain is impermeable with a free slip condition, and the

lower boundary is impermeable with partial slip conditions characterised by a specified drag coefficient of 0.001. The flow is periodic across the lateral boundaries of the domain. The EULAG simulations were calculated for 12:00 UTC $10^{th}$ July 2015 in Ny-Alesund using results obtained from the radiative transfer model and radio-sounding data.

In order to confirm the source region of the BB plume, we applied the Global Environmental Multiscale model with atmospheric chemistry (GEM-AQ; Côté et al., 1998; Kaminski et al., 2008). The GEM-AQ model was run in a global

configuration with a uniform grid resolution of $0.9^{o}$. The vertical domain was defined on 28 hybrid levels with the model top at 10 hPa. Biomass burning emissions were taken from the Global Fire Assimilation System (GFAS; Kaiser et al., 2012). In addition to comprehensive tropospheric chemistry, the GEM-AQ model has 5 size-resolved aerosols species: sea salt, sulphate, black carbon, organic carbon and dust. The microphysical processes that describe formation and transformation of aerosols are calculated by a sectional aerosol module (Gong, 2003). The particle mass is distributed into 12 logarithmically spaced bins from

0.005 to 10.24 µm. The aerosol module accounts for: nucleation, condensation, coagulation, sedimentation and dry deposition,



in-cloud oxidation of $SO_2$, in-cloud scavenging, and below-cloud scavenging by rain and snow. Calculations of $\tau$ are done on-line for all bins and aerosol species. Extinction cross-sections are taken from the AODSEM model (Aubé et al., 2000; Aubé, 2004). The model was run for the period from 15 June to 20 July 2015. Two simulations were carried out - with and without biomass burning emissions. The anthropogenic emissions based on ECLIPSEv4 (http://www.iiasa.ac.at/web/home/research/researchProgra

were used.

## 2.2 Instruments

In this section, we present a brief description of all instruments used for this research studies (Tab. 1). For more detailed specification one is encouraged to read a section concerning instrumentation in Markowicz et al. (2016a), where aerosol optical properties of biomass burning plume in 2015 were taken into consideration.

$\tau$, AE and PW were measured by Full-Automatic Sun Photometer SP1a (Dr. Schulz & Partner GmbH) in Ny-Alesund. The instrument can obtain direct solar radiation in 10 channels ranging from 369 and 1023 nm with 1° of the field of view (Herber et al., 2002). Corrections include temperature variability, Langley methodology and cloud-screening algorithms (Smirnov et al., 2000; Alexandrov et al., 2004).

Extinction profiles were retrieved from KARL Raman lidar. The instrument uses Nd:Yag laser pulse at 355, 532, 1064
nm with the power of 10 W at each wavelength to obtain backscatter and extinction coefficients. Also, the depolarization is measured at water vapour channels (407, 660 nm). The detection is carried out by 70 cm mirror with a 1.75 mrad field of view. Moreover, the overlap issue is fulfilled at 700 m a.g.l. Further details can be found in Hoffmann (2011) and Ritter et al. (2016).

Continuous measurements of radiation fluxes is provided in Ny-Alesund under the Baseline Surface Radiation Network
(BSRN). Ball-shaded CMP22 by Kipp&Zonen installed on solar tracker by Schulz & Partner measure total incoming and reflected solar radiation at 200 - 3600 nm (Maturilli et al., 2015).

The in-situ measurements of single-scattering properties were provided in the Gruvebadet Laboratory, located 1 km south-west from Ny-Alesund. Single wavelength M903 Nephelometer from Radiance Research uses xenon flash lamp and opal diffuser to derive scattering coefficient at 530 nm (Müller et al., 2009) with angular integration range of 10- 170$^o$. Corrections for
non-ideal illumination and truncation error were performed according to the description presented in Müller et al. (2009).

Black carbon (BC) concentration was measured at 467, 530, 660 nm by the means of Particle Soot Absorption Photometer (PSAP) from Radiance Research based on the principle of filter attenuation changes due to aerosol load. Corrections for multiple scattering and non-purely absorbing aerosols were done following methodology from Haywood and Osborne (2000).

Aerosol size distribution measurements were covered by joint spectra of TSI Scanning Mobility Particle Sizer (SMPS 3034)
with 54 channels and TSI Aerodynamic Particle Sizer Spectrometer (APS 3321) with 52 channels jointly in a range of 10-20 000 nm excluding a gap around 500 nm which was fitted. Both instruments delivered data with a resolution of 10 min.





**Table 1.** Description of the instruments installed in Ny-Alesund used as input data for atmospheric radiative transfer model.

| Ground based Instrument | Wavelength, Size [nm] | Quantities* | $\Delta$ t [min] | Station |
|---|---|---|---|---|
| AWI Aerosol Raman lidar KARL | 355, 387, 407, 532, 607, 660, 1064 | $\sigma_{ext}$ | 30 | village |
| AWI Sun photometer SP1a | 369, 381, 413, 500, 610, 674, 779, 860, 945, 1023 | $\tau$, AE, PW | 1 | village |
| Scanning Mobility Particle Sizer Spectrometer SMPS 3034 | 10-487 | ASD | 10 | Gruvebadet laboratory |
| Aerodynamic Particle Sizer APS 3321 | 523-20 000 | ASD | 10 | Gruvebadet laboratory |
| Particle Soot Absorption Photometer | 467, 530, 660 | $\sigma_{abs}$ | 60 | Gruvebadet laboratory |
| Nephelometer M903 | 530 | $\sigma_{scat}$ | 60 | Gruvebadet laboratory |
| Pyranometer | 200-3600 | $F_{in}$, $F_{out}$ | 1 | village |

*$\sigma_{ext}$ - extinction coefficient, $\tau$- aerosol optical depth, AE - Angstrom exponent, PW - precipitable water, ASD - aerosol size distribution, $\sigma_{abs}$ - absorption coefficient, $\sigma_{scat}$ - scattering coefficient, $F_{in}$ - total incoming solar flux, $F_{out}$ - total outgoing solar flux

## 2.3 Inputs to models

### 2.3.1 Vertical profiles of single-scattering properties

The retrieval is based on the in-situ single scattering properties measured at the surface in dry conditions and on vertical profiles of $\sigma_{ext}^a$ as well as RH at ambient conditions from KARL lidar and radio-sounding data. In this study, we assume $\omega^d$

and $R_{eff}^a$ constant with altitude, thus no changes in chemical composition vertically, taking into account the intensity of the BB plume advection. Therefore, by introducing hygroscopic growth model for particles with known size distribution, one may obtain single scattering albedo $\omega$ profile as well as asymmetry parameter $g$ retrieved for ambient conditions. Vertical profiles of single-scattering properties at ambient conditions are used as input parameters to MODTRAN and Monte Carlo calculations.

**Algorithm for delivering single scattering albedo profile $\omega$ at ambient conditions**

From absorption ($\sigma_{abs}$) and scattering ($\sigma_{scat}$) coefficients at 530 nm, $\omega$ can be calculated yielding:

$$\omega(\lambda, z) = 1 - \frac{\sigma_{abs}(\lambda, z)}{\sigma_{ext}(\lambda, z)} \tag{1}$$

at ambient and dry conditions. Subsequently, since $\sigma_{abs}$ is a weak function of RH, the assumption that $\sigma_{abs}^a$ and $\sigma_{abs}^d$ are identical is justified. Then, we can relate dry and ambient conditions by introducing the scattering enhancement factor $f(\lambda, z(RH))$ principle being defined as the ratio between scattering coefficient measured at mentioned RH states:

$$f(\lambda, z(RH)) = \frac{\sigma_{scat}^a(\lambda, z(RH))}{\sigma_{scat}^d(\lambda, z)} \tag{2}$$





where superscripts 'a' and 'd' refer to ambient and dry conditions respectively (Zieger et al., 2010). $f(\lambda, z(RH))$ factor is calculated for 532 nm. Ultimately, we may introduce the equation for $\omega^a$ satisfying:

$$\omega^a(\lambda, z) = \frac{1}{1 + \frac{1 - \omega^d(\lambda, z)}{\omega^d(\lambda, z) \cdot f(\lambda, z(RH))}} \tag{3}$$

Therefore, to derive the relationship between the aerosol water uptake and a particular aerosol species, the Handel model (Hänel, 1976) of growth factor $f(RH)$ is used relating hygroscopicity of aerosols with relative humidity, yielding:

$$f(RH) = \left(\frac{1 - RH^a}{1 - RH^d}\right)^{-\gamma} \tag{4}$$

where $\gamma$ parameter represents the indicator of particles' hygroscopicity. Larger $\gamma$ is referred to more hygroscopic aerosols. In this study, a literature value of $\gamma$ was introduced, equal to 0.18 which applies for biomass burning aerosols (Hänel, 1976). In a range of lidar geometrical compression (0 - 700 m) all data were interpolated from in-situ measurements. The proposed method leads to $\omega^a$ uncertainty of 0.05, where its vast majority may be attributed to $\sigma_{abs}^d$ and $\sigma_{scat}^d$ measurement uncertainties.

**Algorithm for delivering asymmetry parameter $g$ at ambient conditions**

Asymmetry parameter $g$ is derived iteratively using aerosol size distributions measured by SMPS and APS, Mie theory as well as one-parameter equation determined by Petters and Kreidenweis (2007), which approximates relation between RH and growth factor $\chi(RH)$, yielding:

$$\chi(RH) = \left(1 + \kappa \frac{RH}{1 - RH}\right)^{\frac{1}{3}} \tag{5}$$

where RH represents the fraction of relative humidity while neglecting Kelvin effect, which is true for particles affecting significantly light extinction (diameter > 0.01 μm) (Zieger et al., 2011; Bar-Or et al., 2012). Coefficient $\kappa$, however, relates to particle hygroscopicity with respect to Raoult effect. In this study we neglect the effect of broadening of the aerosol size distribution spectra due to diffusional growth of particles for simplification purposes. Carrico et al. (2010) measured $\kappa$ coefficients for different BB aerosols. Regarding HYSPLIT back trajectories for the event in July 2015, Alaskan tundra is acknowledged as a source region. Thus, $\kappa$ coefficient of 0.07 (0.25 μm dry diameter) was chosen to match vegetation (Duff core) covering Alaska. The size distributions of aerosols at ambient conditions were estimated by introducing the hygroscopic growth factor $\chi(RH)$ related to growth of particle due water uptake, yielding:

$$\chi(RH) = \frac{D^a(RH)}{D_d(RH)} \tag{6}$$

where $D$ is the diameter of the particle at the certain RH (Zieger et al., 2010). The calculations are provided for an extreme biomass burning event, thus a concentration of other aerosols than smoke is negligible. That is why, for retrieval of $g$ at ambient conditions by means of Mie theory, we used a constant refractive index for biomass burning aerosol (1.52 - 0.0061i; Sayer et al., 2014).





### 2.3.2 Equations governing 3D Monte Carlo simulations

The relative net irradiance $F_{net}^{rel}$ at the Earth's surface was computed according to the equation:

$$F_{net}^{rel} = \frac{F_{net}}{F_{toa}} = \frac{S_c}{S_s \cdot N_{toa}} \sum_{j=1}^{N} w_j \tag{7}$$

where $RF_{net}$ is the net irradiance aligned with the direction of the vector normal to the sloping surface in column $(k,l)$, $F_{toa}$ is the downward irradiance at the TOA, $N_{toa}$ is the number of photons incident at TOA$(k,l)$, $S_s$ is the area of the Earth's surface within the column $(k,l)$, $S_c$ is the area of the cell $(k,l)$, N is the number of photons absorbed by the Earth's surface within the column $(k,l)$, and $w_j$ is the weight of the $j^{th}$ photon absorbed by the Earth's surface within the column $(k,l)$.

The shortwave direct aerosol radiative forcing (spectral relative radiative forcing) , $RF_{rel}(\lambda)$, is expressed as:

$$F_{rel}(\lambda) = \frac{F_{net}^a(\lambda) - F_{net}^0(\lambda)}{F_{toa}(\lambda)} = F_{net}^{a,rel}(\lambda) - F_{net}^{0,rel}(\lambda) \tag{8}$$

Where superscript '$a$' stands for a clear-sky conditions with an aerosol included and superscript '0' for a clear-sky without one. We can also define RF with respect to the cell surface $S_c$ instead of the actual surface within a given column $S_s$:

$$F_{rel}^{cell}(\lambda) = S_s \cdot RF_{rel}(\lambda) \tag{9}$$

### 2.4 Atmospheric and surface properties - inputs to models

### 2.4.1 Surface properties

$6^{th}$ collection daily product M*D09CMG was used to retrieve surface albedo values over the area between $55^\circ$N and $90^\circ$N with a resolution of $1^\circ$ x $1^\circ$. Data were averaged over the 1 month to obtain a good coverage, thus assumed constant and inserted into Fu-Liou model (Markowicz et al., 2017b).

A model built-in options for the calculations of Fresnel reflection for the oceanic values of surface albedo were performed for the MODTRAN simulations while comparing with Fu-Liou results. Additional set-up of radiometer derived surface albedo was used for the comparison with RF calculated by the means of the radiometer measurements. Both MODTRAN and Fu-Liou codes assume a flat and horizontal Earth's surface.

MODIS MCD43A1 BRDF surface reflectance product for 193 day of 2015 at 469 nm was however used for the 3D Monte Carlo model over Svalbard area. The BRDF was calculated yielding the equation of Strahler et al. (1999):

$$R(\Theta,\vartheta,\phi,\lambda) = f_{iso}(\lambda) + f_{vol}(\lambda) \cdot K_{vol}(\Theta,\vartheta,\phi) + f_{geo}(\lambda) \cdot K_{geo}(\Theta,\vartheta,\phi) \tag{10}$$

where $f$ and $K$ stand for coefficients' kernels, respectively; '$iso$' denotes isotropic scattering component, '$geo$' diffuse reflection component and '$vol$' volume scattering component. $\Theta$, $\vartheta$ and $\phi$ are solar zenith angle, view zenith angle and view-sun relative azimuth angle respectively. Gaps over land were filled with mean values of parameters for a given surface type (glacier or tundra/rock) and elevation range. The coastal line used to distinguish between water and land was taken from the





**Figure 1.** Input maps to the 3D Monte Carlo radiative transfer model: digital elevation model [m] (a), white-sky surface albedo (b), slope aspect (c) and slope inclination [°] (d) retrieved from Norwegian Polar Institute and MODIS products.





Norwegian Polar Institute (2014a). Glaciers outlines (last updated $1^{st}$ April 2016) were taken from Svalbard land covering map dataset (Norwegian Polar Institute, 2014b). Fresnel reflection from the water surface was assumed in the modelling. Moreover, radiation scattering by seawater and its constituents (e.g. phytoplankton or mineral suspended matter) was neglected. The digital elevation model (DEM) used in the Monte Carlo modelling was based on maps from the (Norwegian Polar Institute, 2014a,

UTM zone 33N projection, ellipsoid WGS84). The original DEM was generalized to resolution of 200 m. The Kongsfjord domain covers an area of 91.0 km (x-axis) x 108.0 km (y-axis) and lies among the following coordinates: 78.48°N 10.02°E, 78.52°N 14.11°E, 79.49°N 14.03°E, 79.45°N 9.57°E. The domain consists of cells/columns of 200 x 200 m. The land surface altitude within a cell is estimated by the following equation (Ricchiazzi and Gautier, 1998):

$$z = a_0 \cdot x + a_1 \cdot y + a_2 \cdot x \cdot y + a_3 \tag{11}$$

where $x$ , $y$ and $z$ are the coordinates of a given point of a cell surface and $a_0$, $a_1$, $a_2$ and $a_3$ are coefficients fitted to the coordinates of the cell nodes. The Earth's surface approximated in such a way is continuous. The DEM used in the present modelling is shown in Fig. 1a and white-sky surface albedo in Fig. 1b,c,d show aspect and inclination of the land surface within each cell of the domain. Aspect is azimuth of the vector normal to the surface at the centre of a given cell; inclination is the zenith angle of the vector normal to the surface at the centre of a given cell.

### 2.4.2 Profiles of thermodynamic variables and ozone concentration

Profiles of all thermodynamic properties, including pressure ($p$), temperature, wind speed and relative humidity (RH) were adopted from the radio-soundings performed in Ny-Alesund for the day of the interest. Data above profiles were filled in by subarctic summer profiles of the international standard atmosphere. They were furtherly used for 3D Monte Carlo, MODTRAN and EULAG simulations. The subarctic summer model of ozone profile was used for the simulations after the adjustment to

the real values of ozone measured by SP1a. The profiles for the Fu-Liou calculations were taken from the Navy Operational Global Analysis and Prediction System (NOGAPS). Vertical profiles of ozone were retrieved from dimensional climatology, UGAMP (Li and Shine, 1995) and then scaled to the measured values of the total ozone content by the MODIS M*D09CMG product.

## 3 Results

### 3.1 Retrieval of the single-scattering properties at ambient conditions

Table 2 presents results of the single-scattering properties conversion from dry to ambient conditions described in the 2.3.1 section. 1 - 4 cases represent the time periods where extinction profiles were available. Values of $\omega^d$ at 530 nm (which are assumed constant with altitude) range from 0.92 to 0.95 indicating a moderate absorbing properties characteristic for aged BB plumes. The mean $\omega^d$ obtained for the event is slightly higher than in-situ $\omega^d$ reported by Moroni et al. (2017) which is the

result of the applied additional multiple-scattering correction to PSAP data in this study. Aerosol absorbing properties decrease over the event, resulting in the increase of $\omega^d$ on 11 July to its maximum value of 0.95. The representation of BB plumes



lasting in the atmosphere for more than 3 days in the literature is seldom. Reid et al. (2005) reports a number of mean surface

**Table 2.** Values of retrieved single scattering albedo $\omega$, asymmetry parameter $g$ and effective radius $R_{eff}$ at dry conditions (superscript d) and ambient (superscript a) conditions [530 nm] as well as mean values of variables for the BB period visible in the in-situ measurements.

| Cases [UTC] | $\omega^d(z,550)$ | $g^d(z,550)$ | $g^a(z,550)$ | $R_{eff}^d$ [μm] | $R_{eff}^a$ [μm] | $RH^a$ [%] |
|---|---|---|---|---|---|---|
| 10 Jul 11:30 | 0.92 | 0.61 | 0.62 | 0.17 | 0.18 | 79.0 |
| 10 Jul 23:00 | 0.93 | 0.63 | 0.64 | 0.19 | 0.21 | 83.2 |
| 11 Jul 02:30 | 0.93 | 0.63 | 0.63 | 0.20 | 0.21 | 72.5 |
| 11 Jul 11:30 | 0.95 | 0.59 | 0.60 | 0.10 | 0.12 | 90.0 |
| Mean value | 0.94 | 0.61 | 0.62 | 0.17 | 0.18 | 72.6 |
| (standard dev) | (±0.02) | (±0.02) | (± 0.02) | (±0.02) | (±0.02) | (±0.07) |

$\omega$ ranging from 0.76 to 0.93 characterising aged BB plumes from various in-situ measurements. Although values usually seem to be much lower in comparison to the BB2015 event, the differences result from the definition of aged plumes, where Reid et al. (2005) assumed them to be older than 1 day only. The event is represented by the bimodal aerosol size distribution with a vast majority attributed to the accumulation mode measured at the surface (Markowicz et al., 2016a; Moroni et al., 2017). Although Markowicz et al. (2016a) reports the beginning of the event at 14:00 UTC based on the lidar data we see the temporal discrepancy between in-situ and remote sensing measurements of half a day. Excluding this clearly non-BB period from the in-situ measurements, we obtained 0.17 ± 0.02 μm and 0.18 ± 0.02 μm for effective radius at dry ($R_{eff}^d$) and ambient ($R_{eff}^a$) conditions respectively. Presented results are in good agreement with studies provided by Nikonovas et al. (2015) who reported the values of $R_{eff}^a$ originating from open shrublands to be between 0.176 - 0.194 μm. The obtained asymmetry parameter $g_a$ has values up to 0.62 ± 0.02 that were increased by the conversion from $g_d$ by less than 2 %. All presented variables in Table 1 show an increasing trend throughout the event except for $R_{eff}^d$ and $g_d$ in the last case (11 July 11:30), when possibly the air mass started to change. This trend is likely related to the aerosol ageing processes that caused a reduction of absorbing properties and the particle growth leading to the enhancement of the asymmetry parameter. Vertical profiles of single-scattering properties, as well as relative humidity retrieved at ambient conditions, are presented in Figure 2 that correspond to the time cases presented in Table 2.

The results obtained from the GEM-AQ model simulations confirmed that the BB plume was from wildfires over Alaska. The timing and inflow of aerosol enriched air masses and the rapid increase of $\tau_{550}$ were reproduced correctly in model simulations. Also, vertical profiles of PM10 showed polluted air mass extending up to about 3 km, with maximum concentrations reaching 35 ppb at 2 km. Analysis of 3-D extinction fields over Svalbard shows a thick layer with higher extinction values above the PBL. Modelling results are presented as the $\sigma_{ext}$ profile (Fig. 2b$_{1-4}$). The model reproduced the altitude of elevated extinction coefficients. However, the complex vertical stratification was not captured by the model. On 10 July 11:30 UTC an elevated biomass-burning (BB) layer of enhanced $\sigma_{ext}^a$ derived from the lidar observations is visible between 1 and 3.5 km, where the



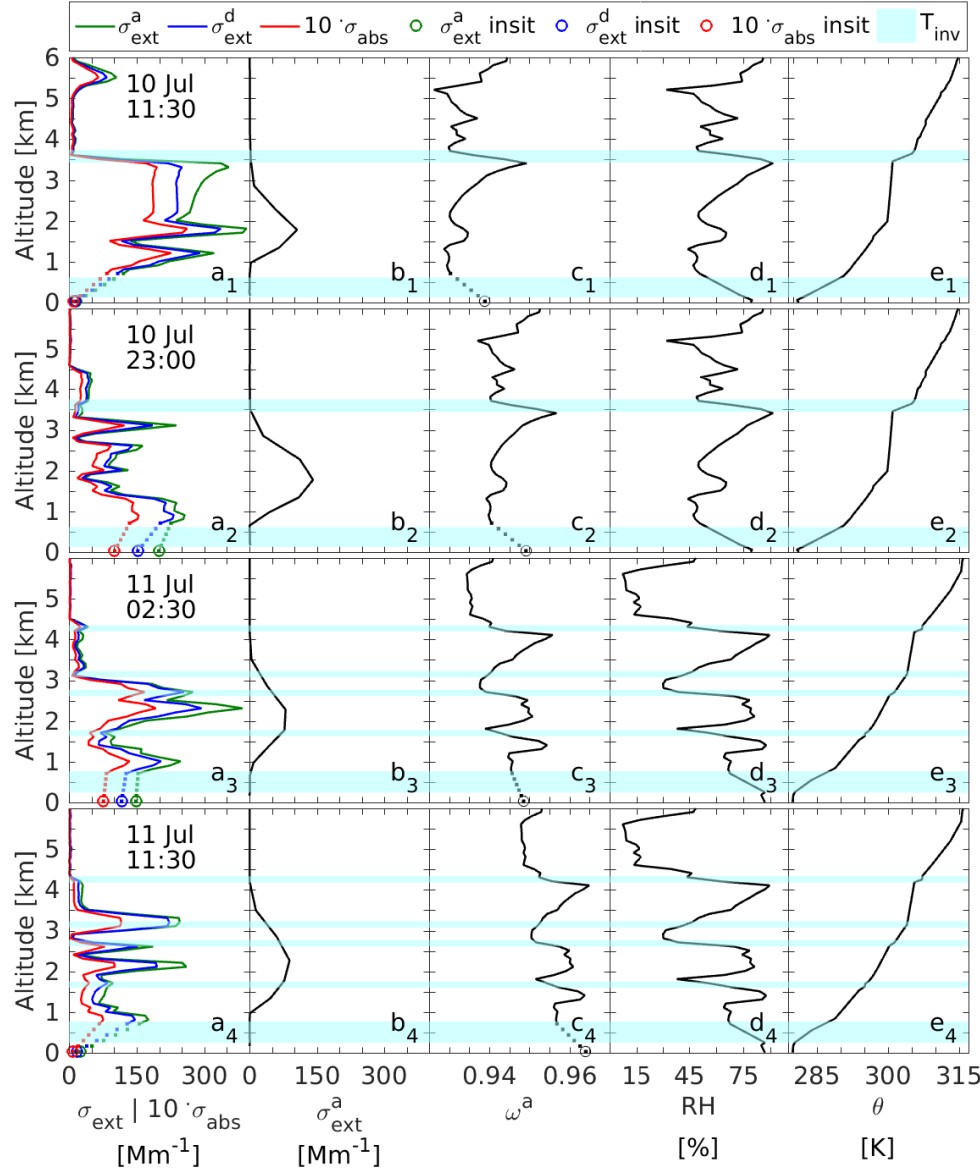

**Figure 2.** Vertical profiles of aerosol single-scattering properties at 530 nm on 10 and 11 of July 2015 (UTC) based on the lidar, radio-sounding profiles and model's output (lines) as well as in-situ measurements (dots). Subfigures include extinction coefficient at ambient (green) and dry (blue) conditions as well as absorption coefficient multiplied 10 times (red; $a_{1-4}$), modelled extinction profile from GEM-AQ ($\sigma_{ext}$; $b_{1-4}$), retrieved single scattering albedo ($c_{1-4}$) at ambient conditions, radio-sounding profiles of relative humidity RH ($d_{1-4}$) and potential temperature $\theta$ ($e_{1-4}$). Blue transparent layers denote temperature inversions ($T_{inv}$).





mean value exceeds 267 $\mathrm{Mm^{-1}}$ (Fig. 2a$_1$). The layer is characterised by three sublayers ( 0.6 - 1.6 km, 1.6 - 2 km and 2 - 3.5 km) which existence is dependent on the temperature inversions ($1^{st}$ and $3^{rd}$) and almost isothermal layer in the middle (Fig. 2e$_1$). The shape of the lower boundary, however, is uncertain due to the overlap issue under 0.7 km of the lidar measurements and therefore lack of the data, which were filled in by the interpolation technique to in-situ values. Additionally, an appreciable

secondary layer is visible around 5.5 km with the mean value of $\sigma_{ext}^a$ around 50 $\mathrm{Mm^{-1}}$ likely connected with thin clouds. The relative humidity obtained from the radio-sounding performed at 12:00 UTC on 10 July varies from 43 to 93 % in the BB layer (Fig. 2d$_1$) altering slightly optical properties of aerosols. Therefore, both $\sigma_{ext}^a$ and RH trend in the layer is positive, while the $\sigma_{ext}^d$ mean value of 225 $\mathrm{Mm^{-1}}$ is much smaller and somewhat constant above 2 km. The average of $\sigma_{abs}$ is 18 $\mathrm{Mm^{-1}}$ and mitigates the shape of $\sigma_{ext}^d$, which results from the retrieval method (eq. 1 and 4). Values of $\omega^a$ profile vary from 0.93 to 0.95,

and the minimum is attributed to the layer around 1.0 km and between 4.0 - 4.5 km (Fig. 2c$_1$). The maximum, however, is observed at the top of the described BB plume and is related to the increase in the RH as a result of our assumptions (eq. 6). In the evening (23:00 UTC), the $\sigma_{ext}$ profiles change significantly indicating the more stratified structure of the BB layer with three main sub-layers (0 - 1.8 km, 1.8 - 2.7 km and 2.7 - 3.2 km) marked out (Fig. 2a$_2$).

The emergence of the lowest one leads to the appearance of the BB particles in in-situ measurements. As indicated by

Markowicz et al. (2016a), in the afternoon, midlevel clouds had formed at around 4.5 km, likely as a result of vertical mixing between BB plume and above air mass. Therefore, a visible additional sublayer at 3.2 - 4.5 might be a possible residuum of their existence. The average value of $\sigma_{ext}^a$ for the BB plume is much lower in comparison to the first profile at 11:30 UTC and equals to 131 $\mathrm{Mm^{-1}}$, while retrieved $\sigma_{ext}^d$ 114 $\mathrm{Mm^{-1}}$ both indicating a deterioration of the upper part of BB plume. Also, $\sigma_{abs}$ decreased significantly reaching the value of 8 $\mathrm{Mm^{-1}}$ in the layer. For the calculations of $\omega^a$ profile, RH profile from the 11

July 12:00 UTC was used, thus small discrepancies between $\sigma_{ext}^a$, $\theta$ and RH profiles are visible. Although the extremal values of $\omega^a$ are similar to those of 11:30 UTC due to assumed retrieval, nevertheless $\omega^a$ ranges between 0.94 and 0.96 indicating the transformation of the BB plume to less absorbing single-scattering properties in comparison to the first case.

The optical conditions on 11 July 02:30 UTC are significantly changed in comparison to the previous case (Fig. 2a$_3$-e$_3$). The average value of $\sigma_{ext}^a$ for BB plume, which is mainly located between 0 and 3.1 km, is slightly higher and exceeds 192 $\mathrm{Mm^{-1}}$.

The plume is divided into 2 sublayers (0 - 1.8 km and 1.8 - 3.1 km) with the enhanced aerosol single-scattering properties mainly located in the middle. Additionally, the layer above BB plume is still visible, as in the previous profile, ranging from 3.1 to 3.5 km and ending with the temperature inversion. Moreover, the hygroscopic properties of aerosols decrease $\sigma_{ext}$ by 18 % while converted to dry conditions. Mean $\sigma_{abs}$, however, is equal to 10 $\mathrm{Mm^{-1}}$ and is similar to the previous case with a $\omega^a$ values ranging from 0.94 to 0.95. This profile was based on the radio-sounding performed at 11 July 12:00 UTC where RH values are

between 33 and 87 % (Fig. 2d$_{3-4}$). The profile mainly consists of three layers with increased relative humidity connected to temperature inversions (0.3 - 0.8, 1.6, 2.7, 4.3 km). Thus, an existence of corresponding $\omega^a$ layers with increased values is visible. As presented in Markowicz et al. (2016a) the lidar range-corrected signal indicates that the last case performed at 11 July 11:30 UTC represents the moment of air mass transformation, possibly altered by low-level cloud formation processes, therefore a significant decay in the optical properties is visible (Fig. 2a$_4$-e$_4$). The residual BB plume is trapped between 0.8 -

3.5 km with no signs of it at the surface (Fig. 2e$_4$). The mean values of $\sigma_{ext}^a$ and $\sigma_{ext}^d$ decreased significantly to 123 $\mathrm{Mm^{-1}}$




and 102 $\mathrm{Mm}^{-1}$ respectively. It consists of complicated sub-layers firmly connected with the existence of layers with stable or neutral conditions (Fig. $2e_4$). Moreover, an average $\sigma_{abs}$ of 5 $\mathrm{Mm}^{-1}$ also revealed a negative trend, however with a smaller magnitude in comparison to $\sigma_{ext}$. The $\omega^a$ values seem to be higher than on July 11 02:30 UTC being between 0.95 and 0.96 within the BB plume, which results from the BB plume transformation.

## 3.2 Temporal variability of radiative forcing on Ny-Alesund

Results presented in this chapter uses $\omega^a$ and $g_a$ retrieval introduced in sections 2.3.1.1 and 2.3.1.2, which has an appreciable influence on the model's outcome. The applied methodology of RF retrieval at ambient conditions indicated the decrease of RF (in magnitude) on average by about 3.1 $\mathrm{Wm}^{-2}$ for the BB event in comparison to the simulations with $g_d$ and $\omega^d$ as inputs. It is due to an increase in both $F_{in}$ and $F_{out}$ by 3.5 $\mathrm{Wm}^{-2}$ and 0.4 $\mathrm{Wm}^{-2}$ respectively for simulation with aerosol included and additional no change in the irradiances from the reference simulation. The impact of the retrieval might be vastly attributed to $\omega$ with the influence of 81% and only 19 % to g.

Figure 3 presents the comparison of irradiances (Fig. 3a) and clear-sky RF (Fig. 3b) obtained by the means of MODTRAN simulations and estimated both by the radiometers' measurements in Ny-Alesund and model calculations (reference case) for the BB2015 event. The latter represent all-sky conditions since the discussed BB event is extremely complicated and therefore a possible cloud contamination seems to be impossible to separate entirely. However, periods with a clear influence of clouds were removed, therefore presented mean value of RF lacks most intense period (see Fig. 3b). For these calculations a real value of broadband albedo was used. The solid black line in Fig. 3a shows a distribution of total incoming irradiances ($F_{in}$) measured by radiometers and black dots symbolise modelled $F_{in}$ for the case with aerosol load included, blue dots, however, represent modelled $F_{in}$ for the case free of aerosols $F_{cin}$ (reference simulation). The daily variability of the latter is mainly the function of solar zenith angle and for the 9 - $11^{th}$ of July 2015 ranges from around 153.0 $\mathrm{Wm}^{-2}$ at midnight to 560.8 $\mathrm{Wm}^{-2}$ at noontime, $F_{in}$, on the other hand, is additionally strongly affected by the optical and physical properties of the advected smoke. The model's performance at background conditions might be validated at the period between 7:00 to 14:00 UTC on $9^{th}$ July. It represents the clear-sky period with an infinitesimal load of aerosols, typical for summer background conditions in the Arctic. Both measured and modelled $F_{in}$ are in a rather good agreement deviating on average only by 9.7 $\mathrm{Wm}^{-2}$ (2 %) from each other and additionally by 0.4 % as well as 2.3 % from the reference simulation. Measured and modelled $F_{out}$ indicates a very good agreement of less than 1 %, which values exceeds 69.8 $\mathrm{Wm}^{-2}$ (Radiometers: later on as Rad) and 69.4 $\mathrm{Wm}^{-2}$ (MODTRAN hereinafter referred to as Mod) respectively. At 14:00 UTC Markowicz et al. (2016a) reports an advection of BB plume on average located between around the surface and 3.5 $\mathrm{km}$, as indicated by ceilometer and lidar measurements. From the latter, one might conclude a complicated structure of the BB layers with a mixture of aerosol and clouds included, which lasts over Ny-Alesund until around noon on $11^{th}$ July. The mean values of modelled $F_{in}$ during the event might be estimated at the level of 243.0 $\mathrm{Wm}^{-2}$ while Mod $F_{cin}$ of 332.1 $\mathrm{Wm}^{-2}$. Therefore, the existence of the BB aerosol reduced the incoming solar irradiance by around 90 $\mathrm{Wm}^{-2}$ when compared to the reference case. Furthermore, we report the average value of Mod $F_{cout}$ reaching 47.1 $\mathrm{Wm}^{-2}$ and Mod $F_{out}$ at the level of 36.9 $\mathrm{Wm}^{-2}$. In overall, performed simulations of the BB event by means of MODTRAN model usually leads to underestimation of $F_{in}$ and overestimation of $F_{out}$ while compared to the




measurements provided by radiometers. This result needs to be interpreted with caution as radiometer data might be partially cloud contaminated. The estimated accuracy, however, is around 10 % and 5.8 % respectively. The highest decrease in Mod $F_{in}$ is visible on $10^{th}$ July, where reduction exceeded 30 % in comparison to daily Mod $F_{cin}$ and reached 27 % regarding the summer background conditions (7:00 - 14:00 UTC on $9^{th}$ and $10^{th}$ July). It is believed to be indicated by the observed

maximum of $\tau_{550}$. Additionally, higher variability of Rad $F_{in}$ is visible in comparison to the $9^{th}$ July, which might be the result of increased turbulence and a possible aerosol activation (see section 3.4). Further, a number of high- and mid-level clouds are reported around noon and in the afternoon (Markowicz et al., 2016a) - thus, radiometer data are removed. We report

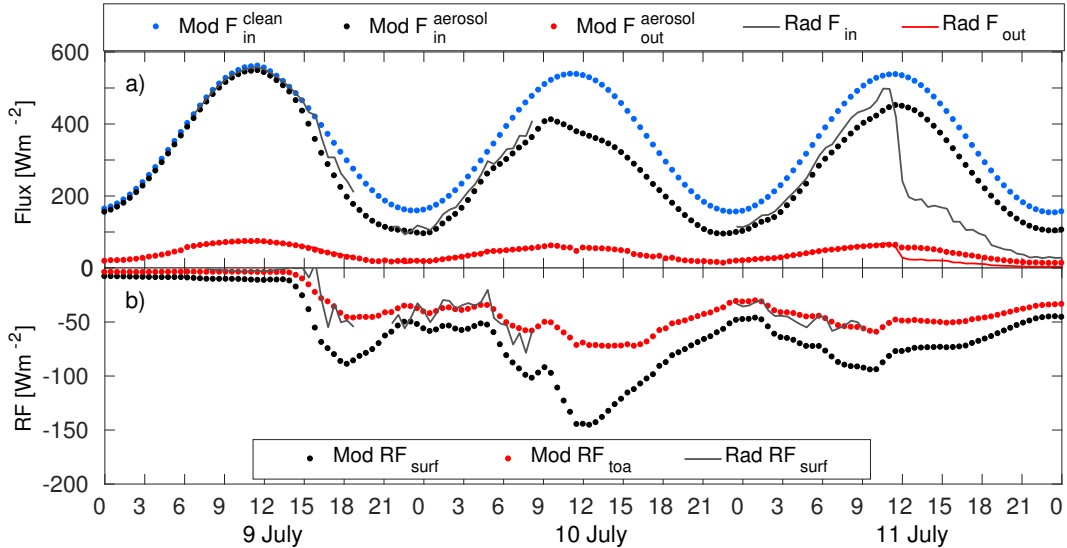

**Figure 3.** The mean daily values of radiative forcing ($RF$) calculated by means of Fu-Liou (FuLiou) and MODTRAN (Mod) models. Simulations were run for clear-sky conditions at the surface (subscript surf), within the atmosphere (subscript atm) and at the top of atmosphere (subscript toa). The surface reflectance in MODTRAN simulations is based on the Fresnel reflection calculations at the ocean surface.

the average $RF_{surf}$ for the studied smoke plume on the level of -78.9 Wm$^{-2}$ (Mod) and -43.3 Wm$^{-2}$ ( Rad) indicating a significant cooling effect of BB aerosol at the surface. The highest values (in magnitude) are observed at around 12:00 UTC

on $10^{th}$ July, being attributed to, as previously mentioned in this chapter, highest values of $\tau_{550}$. Thus, a momentary Mod $RF_{surf}$ exceeded -147 Wm$^{-2}$ regarding MODTRAN simulations. Similar results were reported by Stone et al. (2008), who studied smoke advected from Alaska to the Canadian Arctic during $2^{nd}$ July 2004. Authors came to conclusions that an average diurnal $\tau_{500}$, of 0.5 would produce a cooling effect at the surface reaching -40 Wm$^{-2}$. Since in our case the average $\tau_{550}$ is 0.64 the results seem to be complementary. On the other hand, a study from Sitnov et al. (2013) reveals smaller absolute values

of $RF_{surf}$ at much higher $\tau_{550}$ for the wildfires observed in the European Russia at the beginning of August 2010. For the average $\tau_{550}$ being between 0.98 - 1.16 authors estimated $RF_{surf}$ to be around -60 Wm$^{-2}$. As $RF_{surf}$ is a function of both solar zenith angle (Stone et al., 2008) and surface albedo (Carslaw et al., 2010) that might explain the reported differences.





The average value of Mod $RF_{toa}$ exceeded -47 Wm$^{-2}$ indicating BB plume to cool the entire atmospheric column. Within the atmosphere, it has, however, a positive impact of 31.9 Wm$^{-2}$ (Mod $RF_{atm}$). This pattern is in agreement with Stone et al. (2008) who also reports negative values at TOA and a warming when an atmospheric layer is considered. In overall, the described plume had about 31 % higher (in magnitude) influence at the surface in comparison to TOA. Model calculations
usually overestimate Mod $RF_{surf}$ values, which on average deviate from Rad $RF_{surf}$ by around 32.9 %, possibly related to all-sky conditions which increase diffusive flux. This result should be additionally interpreted with caution since $RF_{surf}$ from radiometers might be cloud contaminated and cause increased variability of the obtained quantity.

High single-scattering albedo values and negative $RF_{toa}$ clearly show that scattering is dominant with respect to the light absorption contribution. Indeed absorption species (mainly BC) are able to mitigate the cooling effect of the BB event into the
atmosphere, but not sufficient to change the RF sign. This means that BC particles play a minor role with respect to scattering particles (sulfate, OC, etc.). That could be demonstrated also by the changes in atmospheric concentrations of BC, OC, sulfate and oxalate measured at Gruvebadet. In particular, the relative concentrations increase of about 20 times for BC (and EC), OC and oxalate, and about 10 times for non-sea-salt sulfate during the BB event, with respect to the background level. In spite of the BC and OC relative increases are similar, the absolute concentrations of OC are significantly higher than BC (and EC),
reaching values as high as 4500 ngm$^{-3}$ i.e. more than 10 times higher than atmospheric concentration of BC (around 300 ngm$^{-3}$; Moroni et al., 2017).

Although RFs reported in our study agree with other results, we believe that $RFE_{surf}$ is more accurate quantity to intercompare with other studies regarding intrinsic aerosol properties. The mean Mod $RFE_{surf}$ value of the BB event in Svalbard of -126 Wm$^{-2}$/$\tau_{550}$ is slightly higher in comparison to other estimates of smoke transport like -100 Wm$^{-2}$/$\tau_{550}$
reported by Markowicz et al. (2016b) for the Canadian forest fires advection over Europe in 2013 and -90 Wm$^{-2}$/$\tau_{550}$ for wildfires observed over Create in 2001 (Markowicz et al., 2002). On the other hand, multiyear mean $RFE_{surf}$ values obtained for different regions are appreciably higher, i.e. $RFE_{surf}$ from tropical forest wildfires is estimated over Amazon basin on the level from -140 ± 33 Wm$^{-2}$/$\tau_{550}$ through boreal forest fires from North America of -173 ± 60 Wm$^{-2}$/$\tau_{550}$ and ending with $RFE_{surf}$ for African Savannas at the level of -183 ± 31 Wm$^{-2}$/$\tau_{550}$ (García et al., 2012). The reported differences
might be explained by discrepancies in solar zenith angle and intensive aerosol properties. In particular, absorbing properties are a function of black carbon concentration and the latter in turn is mostly produced during flaming combustion. Flaming combustion plays a crucial role regarding African Savannah (90 %) and lesser in Amazon and Boreal forests of 50 % and below 20 % respectively. Therefore $\omega$ equals to 0.85 ± 0.04, 0.91 ± 0.04 and 0.95 ± 0.04 respectively (García et al., 2012, and references therein). It might be especially important in case of African and Amazonian regions.

**3.3   The comparison of RF derived from MODTRAN and Fu-Liou simulations**

This chapter focuses on the comparison of derived RFs between DISORT radiative transfer code and much more robust $\delta$ 2/4 stream approximation, represented by MODTRAN and Fu-Liou models respectively. The results of the latter were previously published in Markowicz et al. (2017b) regarding BB transport over the Northern Hemisphere. In this study, all RFs and RFEs were retrieved over ocean area near Ny-Alesund (78.5$^\circ$N, 9.5$^\circ$E) to neglect albedo variability of the land




surface in this comparison and they were compared with MODTRAN simulations which assumed spectral dependence of the Fresnel reflection over a water body. Table 3 presents the comparison between input variables to both models: mean daily $\omega^a$, precipitable water (PW) and $\tau_{550}$. Column-integrated $\omega^a$ is calculated yielding (Schafer et al., 2014):

$$\omega^a = \frac{\int\limits_0^{10km} \sigma_{ext}^a(z) \cdot \omega^a(z) dz}{\tau} \tag{12}$$

While $\omega^a$ in case of MODTRAN simulations has an increasing trend within 9 - 11$^{th}$ July from 0.92 to 0.96, the same quantity show more absorbing properties by 3 - 6 % and is rather constant for Fu-Liou calculations oscillating around 0.91 - 0.93. The same trend is visible for PW mean values, where it is between 1.72 - 2.26 cm for MODTRAN simulations and Fu-Liou it is by 10 - 40 % lower ranging from 0.98 to 2.08 cm. Additionally, retrieved $\tau_{550}$ from SP1a measurements equal to 0.23 - 0.72 and NAAPS calculations at 1$^o$ x 1$^o$ resolution of 0.2 - 0.59, seem to deviate from each other by 8 - 35 %. What is more, while

the highest $\tau_{550}$ value for MODTRAN simulations is on 10$^{th}$, for the Fu-Liou, however, it is noticeable on 11$^{th}$. Presented discrepancies between variables are satisfactory given the fact of retrieving them over much larger spatial resolution in case of Fu-Liou model. Figure 4 presents the daily mean values of RFs derived from MODTRAN and Fu-Liou calculations for

**Table 3.** The mean daily values of the single-scattering albedo $\omega^a$, precipitable water PW [cm] and aerosol optical depth $\tau_{550}$ at 550 nm used as inputs to MODTRAN and Fu-Liou simulations.

|  | $\omega^a$ | | | PW [cm] | | | $\tau_{550}$ | | |
|---|---|---|---|---|---|---|---|---|---|
|  | 9$^{th}$ | 10$^{th}$ | 11$^{th}$ | 9$^{th}$ | 10$^{th}$ | 11$^{th}$ | 9$^{th}$ | 10$^{th}$ | 11$^{th}$ |
| Modtran | 0.92 | 0.94 | 0.96 | 1.72 | 2.26 | 2.22 | 0.23 | 0.72 | 0.55 |
| FuLiou | 0.93 | 0.91 | 0.92 | 0.98 | 2.08 | 1.98 | 0.20 | 0.54 | 0.59 |

BB2015 event at the surface, atmosphere ($RF_{atm}$) as well as at TOA for clear-sky conditions. On 9$^{th}$ July a mean $RF_{surf}$ is -29.5 Wm$^{-2}$ and -20.8 Wm$^{-2}$ regarding MODTRAN (blue bar with turquoise edges) and Fu-Liou (grey bar with black edges)

simulations. The radiative effect of BB plume in the atmosphere ($RFs_{atm}$; a light yellow bar with a red edge [MODTRAN] as well as a light blue bar with dark blue edges [Fu-Liou]) seem to be much lower in magnitude when compared to $RF_{surf}$, however with the opposite sign indicating local heating effect. $RFs_{atm}$ exceed values of 15.0 Wm$^{-2}$ and 8.5 Wm$^{-2}$ for MODTRAN and Fu-Liou calculations respectively. $RF_{toa}$ (yellow bar with brown edges and green bar with dark green edges) is characterised by a cooling effect of aerosols on the level of -14.6 Wm$^{-2}$ and -12.3 Wm$^{-2}$ regarding MODTRAN as well

as Fu-Liou simulations. 10$^{th}$ July is characteristic for the highest levels (in magnitude) of almost all presented daily mean RFs. $RFs_{surf}$ exceeds values of -88.1 Wm$^{-2}$ (MODTRAN) and -76.9 Wm$^{-2}$ (Fu-Liou). Mean $RFs_{atm}$, however, increased by more than 300 % in comparison to the previous day and were of 35.9 Wm$^{-2}$ and 35.8 Wm$^{-2}$ respectively. $RF_{toa}$ also intensified by a factor of 0.6 - 3 to reach -52.2 Wm$^{-2}$ and -41.2 Wm$^{-2}$ regarding MODTRAN and Fu-Liou simulations. Although highest $\tau_{550}$ values are reported on 11$^{th}$ for the latter - RFs maximum is observed on 10$^{th}$. $\tau_{550}$ effect, in this case,




is believed to be enhanced by lower $\omega^a$. Given the fact that $RF_{toa}$ for all-sky conditions modelled by Fu-Liou is equal to -14.0 $\mathrm{Wm}^{-2}$ (not shown) on $10^{th}$, these results are considered exceptional in the Arctic records and are of similar magnitude as other investigation on aerosol high load event in this region. Thus, all-sky $RF_{toa}$ for BB transport in 2006 was estimated at the level of between -12 and 0 $\mathrm{Wm}^{-2}$ (Myhre et al., 2007). In the last day of the measured BB event over Ny-Alesund

RFs slightly decreased in comparison to the $10^{th}$ July. However, they were still significantly high when compared to the calculated mean impact of absorbing aerosols in the summertime for the Arctic region of -1.2 $\mathrm{Wm}^{-2}$ at the surface and -0.19 $\mathrm{Wm}^{-2}$ at TOA (Breider et al., 2017). $RF_{surf}$ reached levels of -67.6 $\mathrm{Wm}^{-2}$ and -71.0 $\mathrm{Wm}^{-2}$ respectively. One might see a noticeable inversion of a relation between clear-sky MODTRAN and Fu-Liou simulations where the latter was usually lower (in magnitude) in comparison to MODTRAN, and it might be related both to a significant difference in the mean $\omega^a$ as well

as $\tau_{550}$ as indicated in table 2. The same situation is visible for $RFs_{atm}$, which on $11^{th}$ July were equal to 23.7 $\mathrm{Wm}^{-2}$ and 31.2 $\mathrm{Wm}^{-2}$ respectively, while $RFs_{toa}$ exceeded values of -43.9 $\mathrm{Wm}^{-2}$ as well as -39.8 $\mathrm{Wm}^{-2}$ for both simulations. In

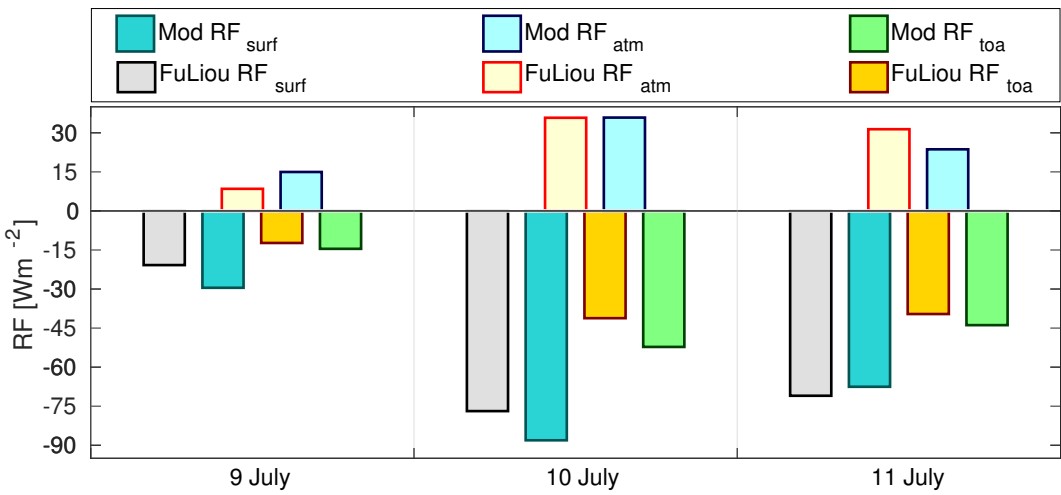

**Figure 4.** The mean daily values of radiative forcing ($RF$) calculated by means of Fu-Liou (FuLiou) and MODTRAN (Mod) models. Simulations were run for clear-sky conditions at the surface (subscript surf), within the atmosphere (subscript atm) and at the top of atmosphere (subscript toa). The surface reflectance in MODTRAN simulations is based on the Fresnel reflection calculations at the ocean surface.

overall, the difference between daily mean values of MODTRAN and Fu-Liou simulations is close on average to around 15 % with all assumed input variables being lower for the latter. Since for each model, different resolution of input parameters over the slightly distinct area was used, authors consider obtained accuracy fairly good. Differences between Modtran and Fu-Liou

simulations are vastly connected with a slightly different aerosol optical properties.

Aerosol direct radiative forcing efficiency (RFE) defined in this paper as the ratio between RF and $\tau_{550}$ is presented in Figure 5 calculated for MODTRAN and Fu-Liou simulations. The highest values of RFEs (in magnitude) were observed at the surface ($RFE_{surf}$) for the event, ranging from -105.8 $\mathrm{Wm}^{-2}/\tau_{550}$ on $11^{th}$ July up to -159.9 $\mathrm{Wm}^{-2}/\tau_{550}$ on $9^{th}$ regarding



MODTRAN simulations as well as from -104.0 Wm$^{-2}$/$\tau_{550}$ on $9^{th}$ July to -142.4 Wm$^{-2}$/$\tau_{550}$ on $10^{th}$ July according to Fu-Liou model. The $RFE_{atm}$ is between 39.6 and 71.2 Wm$^{-2}$/$\tau_{550}$ on $9^{th}$ July regarding MODTRAN simulations, while Fu-Liou results show its values being on average higher ranging from 42.5 - 66.3 Wm$^{-2}$/$\tau_{550}$. $RFE_{toa}$ varies from -71.0 Wm$^{-2}$/$\tau_{550}$ to -86.6 Wm$^{-2}$/$\tau_{550}$ for MODTRAN calculations as well as from -61.5 Wm$^{-2}$/$\tau_{550}$ to -76.3 Wm$^{-2}$/$\tau_{550}$ for Fu-Liou model.

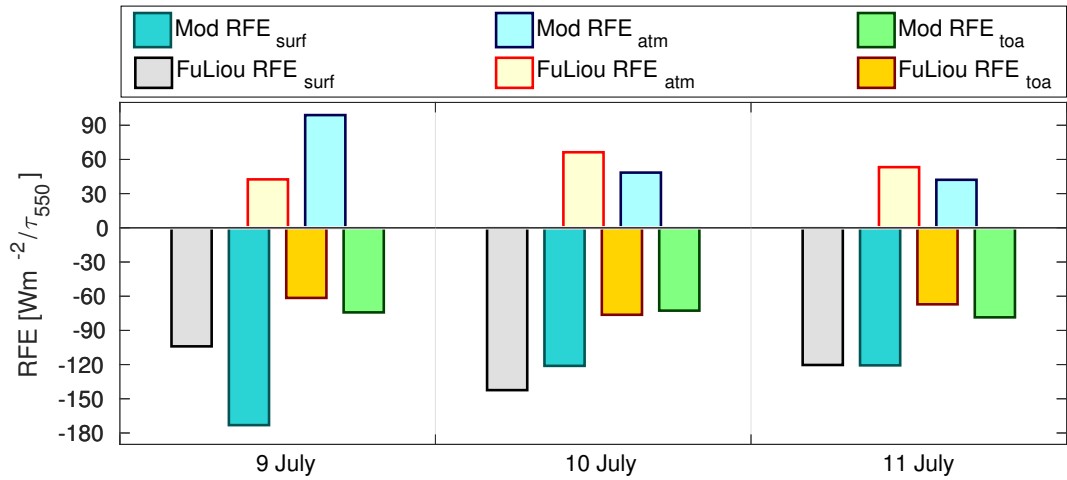

**Figure 5.** The mean daily values of radiative forcing efficiency (RFE) calculated by means of Fu-Liou (FuLiou) and MODTRAN (Mod) models. Simulations were run for clear-sky conditions at the surface (subscript surf), within the atmosphere (subscript atm) and at the top of atmosphere (subscript toa). The surface reflectance in MODTRAN simulations is based on the Fresnel reflection calculations at the ocean surface.

## 3.4 3D distribution of radiative forcing in the vicinity of Ny-Alesund valley

In the previous sections, we discussed the RF computed for a single cell. In this approach, both topographic effects (shading, slope inclination, etc.) and small (subgrid) scale variability in surface albedo were neglected. Moreover, photon transfer between atmospheric columns was assumed zero. In the present section, we use 3D geometry and 3D Monte Carlo simulations

10 of radiative transfer to show an example of the variability of the RF in the vicinity of Ny-Alesund. The simulations were performed for a single wavelength $\lambda$=469 nm and the solar position for the time of the retrieval of aerosol properties' profile ($10^{th}$ July 2015 11:30 UTC; solar zenith angle=57$^{\mathrm{o}}$, solar azimuth=173$^{\mathrm{o}}$). RF was also computed using the plane-parallel geometry for individual 200 m cells/columns (ICA - Independent Column Approximation). In this section, RF is expressed as a fraction of downward irradiance at toa (Eq. 7-9). Further in this section, we will skip $\lambda$ and $RF_{rel}$, $RF_{rel}^{cell}$ and $RF_{rel}^{pp}$ will

15 denote relative spectral RF simulated using the 3D modelling, $RF_{rel}(\lambda$=469 nm), $RF_{rel}^{cell}(\lambda$=469 nm), and the plane-parallel approach to individual cells, $RF_{rel}^{pp}(\lambda$=469 nm).

Figure 6 shows the spatial distribution of $RF_{rel}$ (Eq. 8) and compares it to the distribution derived using the ICA approach. The mean values of RF and the standard deviations are compared in Table 4. In the analysed case the domain mean values





and standard deviation of $RF_{rel}$ is -0.1817 ± 0.1066 for the RF calculated with respect to the real inclined surface (i.e. per unit area of the inclined surface; compare Eq. 7-8), and $RF_{rel}^{cell}$ is -0.1875 ± 0.1104 when the RF is calculated with respect to the horizontal cell surface (i.e. per unit area of the cell surface; compare Eq. 9). There is a large difference between the RF over water and land surfaces, which is mainly due to differences in surface albedo between these regions. For the fjord

5 surface, an absolute value of RF is smaller and weakly variable over the fjord surface, mean $RF_{rel}^{cell}$ is equal to mean $RF_{rel}$ and reaches -0.2632 ± 0.0092. Its coefficient of variation is 3.5%. The actual value of RF variability over the sea may be even lower because the noise of Monte Carlo method may enhance it. The land RF is characterised with both $RF_{rel}^{cell}$ and $RF_{rel}$ less negative mean values of -0.1395 ± 0.1180 as well as -0.1326 ± 0.1084, respectively, and much stronger surface variability. The respective coefficients of variation are 84.6% and 81.7%. In our simulation, the variability of RF over the sea is caused

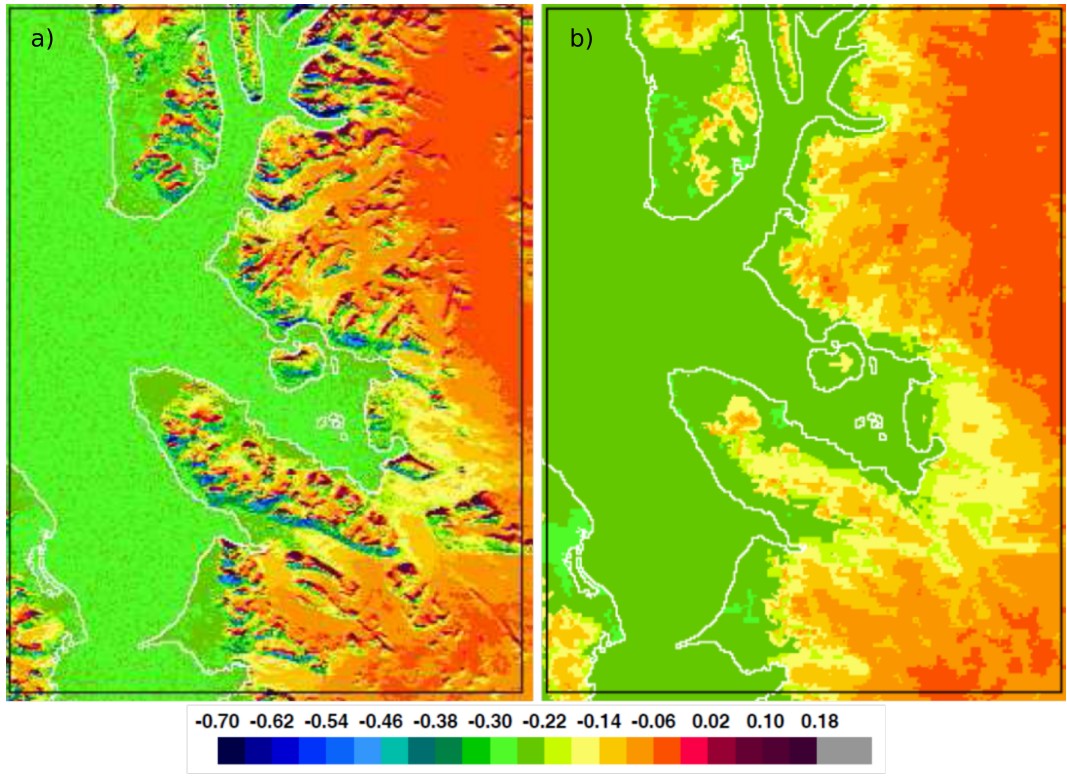

**Figure 6.** A comparison of the $RF_{rel}$ distribution derived from 3D Monte Carlo model with $RF_{rel}^{pp}$ (a) at 469 nm and computed applying plane-parallel geometry of the Monte Carlo model to each pixel independently (ICA) (b). Computations for $\lambda$=469 nm, solar zenith angle=57°, solar azimuth=173°, and aerosol properties of 10 July 2015, 11:30 UTC.

10 by an impact of the surrounding land only. Apart from shading the sky and the sun by the orography, the spatial variability of RF and its deviations from the plane-parallel RF values are caused by positive net horizontal photon transfer from the land area. Horizontal photon transfer due to reflection between the atmosphere and the underlying surface is efficient over bright areas like snow-covered land and glaciers. The horizontal distance of the photon transmission outside the bright underlying



surface relates to the effective height at which the radiation reflected upward by the Earth's surface is reflected downward by the atmosphere. The net horizontal transport is observed for both atmospheres, with and without aerosols, but in each case the effective height of reflection is different. An appearance of dense low-lying aerosol layer reduces the effective reflection height and thus horizontal distance the photons can travel over the fjord, but at the same time, it intensifies the reflectance of

the atmosphere comparing to the case without aerosols. Thus the gradient in irradiance with distance from the reflective land is stronger in the aerosol case. The atmosphere without aerosols acts similarly to a very thin cloud located higher over the Earth's surface while aerosol layer can be compared to a thicker cloud with its base at a lower height (Rozwadowska and Górecka, 2012).

The main factors influencing RF and its variability over land in the vicinity of Ny-Alesund comprise reflective properties of

the land surface, slope exposition concerning the sun and shading of the sun by the mountains. The impact of photons reflected from nearby sunlit slopes and horizontal photon transport due to multiple reflections between the sky and the surface on RF variability are of secondary importance over the land.

In the analysed case the highest magnitude of negative RF was found for sun-facing slopes of white sky albedo of around 0.2. In such places, the effective solar zenith angle is relatively low and high contribution of the direct solar radiation to the total

irradiance results in a substantial reduction in the surface irradiance due to the presence of aerosols, and thus in $RF_{rel}$ of about -0.39. For slopes lit by diffused radiation mainly, the RF is positive, i.e. presence of aerosols increases the amount of radiation absorbed by the surface. In shaded places with the effective solar zenith angle of about $90^o$ and white sky albedo of around 0.4 $RF_{rel}$ can be as high as 0.07 in our simulation. Using the ICA approach to RF estimation results in an underestimation

**Table 4.** Mean relative radiative forcing RF calculated concerning the actual surface, $RF_{rel}$, and the horizontal cell surface $RF_{rel}^{cell}$ using the 3D Monte Carlo model. $RF_{rel}^{pp}$ is RF computed using the Independent Column Approximation approach. Computations were done for $\lambda$=469 nm, solar zenith angle=$57^o$, solar azimuth=$173^o$, and aerosol properties of the $191^{st}$ day of 2015.

|  | All cells | Water | Land |
| --- | --- | --- | --- |
| $RF_{rel}^{cell}$ | -0.1875 ± 0.1104 | -0.2632 ± 0.0092 | -0.1395 ± 0.1180 |
| $RF_{rel}$ | -0.1817 ± 0.1066 | -0.2632 ± 0.0092 | -0.1326 ± 0.1084 |
| $RF_{rel}^{pp}$ | -0.1842 ± 0.0824 | -0.2586 ± 0.0 | -0.1372 ± 0.0734 |
| $RF_{rel}^{cell}$-$RF_{rel}^{pp}$ | -0.0032 ± 0.0699 | -0.0047 ± 0.0092 | -0.0024 ± 0.0890 |

of the surface variability in the RF. It also results in biased domain mean values of the RF. In the case under study, the mean

difference between the more accurate RF for the horizontal cell surface and the RF calculated using the plane-parallel approach, $RF_{rel}^{cell}$ and $RF_{rel}^{pp}$ are -0.0032 ± 0.0699, which is 1.9 % of the mean $RF_{rel}^{cell}$. This, in conversion to daily mean shortwave RF, gives the average error not exceeding 2 $Wm^{-2}$ while using plane-parallel approach. Thus, it is almost as high as the effect of $\omega^d$ translation to ambient conditions considered in our study. Additionally, the mean bias is higher for the sea than for the land. However, for individual cells/columns, the variability of deviations from the real value of $RF_{rel}^{cell}$ is much larger for




the land where the standard deviation of the difference $RF_{rel}^{cell}$-$RF_{rel}^{pp}$ equals 63.8% of the mean $RF_{rel}^{cell}$. The negative bias with the largest magnitude, 0.247, was found for the case of sun-facing slopes discussed above. For shaded inclined areas, the plane-parallel approach seriously underestimates radiative forcing, the mean bias equals 0.233.

### 3.5 Impact of biomass burning aerosol on the atmospheric dynamics

5    LESs performed using the EULAG model indicate an appreciable impact of the BB plume on atmospheric dynamics. Figure 7 presents the development of potential temperature and turbulent kinetic energy (TKE) in a reference simulation (Fig. 7b,c) representing a clear atmosphere and in a polluted simulation (Fig. 7e,f) including effects related to the BB plume. Initial profiles used in the simulations are based on the radio-sounding from $10^{th}$ July 12:00 UTC, and the applied heating rates, given by

$$r_h = \frac{1}{\rho \cdot C} \frac{\partial F_{net}}{\partial z} \qquad (13)$$

where $\rho$ is air density and $C$ is a specific heat capacity defined both for short- and longwave irradiances obtained from

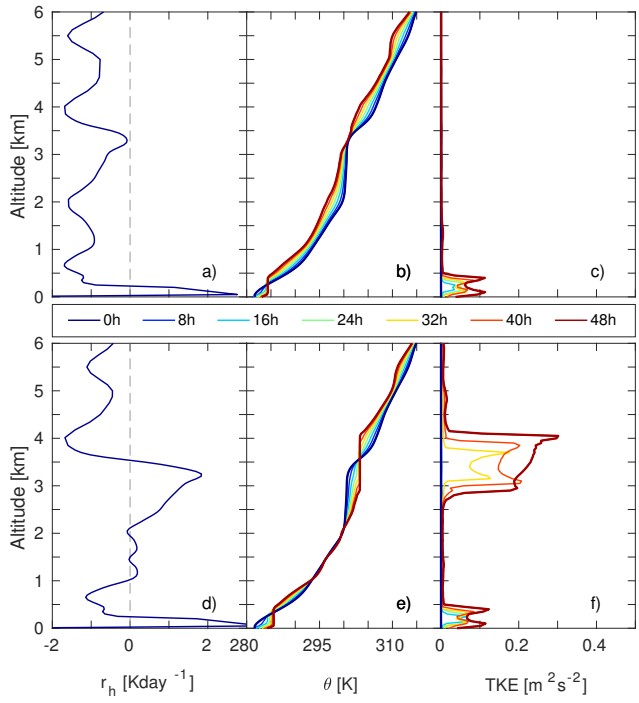

**Figure 7.** Horizontally averaged profiles of heating rate $r_h$ (a,d), potential temperature $\theta$ (b,e), and turbulent kinetic energy TKE (c,f) for simulations of a reference case (a-c) and a polluted case with effects of aerosol load included (d-f). The performed EULAG simulations are based on profiles retrieved $10^{th}$ July 2015 11:30 UTC and simulation data is stored at 8 h intervals.

MODTRAN simulations for $10^{th}$ July 11:30 UTC. The $r_h$ profiles for the reference case (Fig. 7a) and the aerosol polluted case (Fig. 7d) both show a thin layer near the surface ($z$<0.5 km with significant heating; 2.7 and 3.4 Kday$^{-1}$ respectively. Above 0.5 km, the reference case indicates the cooling of the atmosphere at a rate of approximately of 1 Kday$^{-1}$, while





in the polluted case, another layer with significant heating effects is visible between altitudes of 1 and 3.5 km. The heating rate in the lower part of this layer is around 0.2 $\mathrm{Kday}^{-1}$ while in the upper part it reaches values of up to 1.8 $\mathrm{Kday}^{-1}$. The two simulations have the same initial profile of $\theta$ which is represented by the navy blue lines in Figure 7b,e. There is a layer between altitudes of 2 and 3 km with nearly constant initial $\theta$, but in general, it decreases with altitude. Due to the stable initial stratification and the lack of, e.g. strong surface heating, turbulence develops slowly in the performed simulations (see TKE profiles in Fig. 7c,f). After 16 h, turbulent layers start to develop near the surface in both simulations. The TKE in these layers reaches values of around 0.1 $\mathrm{m^2s^{-2}}$, and they extend up to 0.5 km at the time $t=48$ h. After 24 h, a second turbulent layer starts to develop in the polluted case at an altitude of approximately 3.4 km. The thickness of this layer increases with time, and at $t=48$ h it covers altitudes between 2.5 and 4.2 km with maximum TKE values of 0.3 $\mathrm{m^2s^{-2}}$ and updrafts/downdrafts with vertical velocities of around 1 $\mathrm{ms^{-1}}$. In contrast, the flow in the reference case remains almost non-turbulent above 0.5 km with vertical velocities close to zero throughout the simulation period. In the regions with relatively high TKE, $\theta$ becomes nearly constant with altitude, and the polluted simulation indicates that the initially well-mixed layer around $z=2.5$ km expands and moves upwards over time. Outside the clearly turbulent regions, very little vertical mixing takes place, and the potential temperature is approximately given by:

$$\theta = \theta(0,z) + r_h \cdot t \tag{14}$$

where '$z$' symbolises altitude and '$t$' time.

## 4 Conclusions

This paper presented the investigation of the BB aerosol high-load event that was observed during $9^{th}$-$11^{th}$ July 2015 in the Arctic. In this investigation, we focused on the local perturbations in the radiation budget as well as atmospheric dynamics concerning Ny-Alesund area on Spitsbergen. The discussed BB advection was one of the most spectacular regarding last 25 years (Myhre et al., 2007) with all aerosol optical properties typical for the summer conditions enhanced by the factor of more than 10. In particular, mean daily values of $\tau_{550}$, PW and $\omega$ exceeded 0.2-0.7, 1.7-2.2 cm as well as 0.93-0.97 respectively according to in-situ and photometer data at Ny-Alesund. Here, we wanted to underline the most significant outcome from the paper:

– Simulations with the GEM-AQ model confirmed the source region and the arrival time at Ny-Alesund of the Alaskan BB plume indicating a reasonable agreement in the profile of $\sigma_{ext}$. The apparent underestimation of aerosol loading in the plume might be connected to a rather coarse horizontal and vertical resolutions.

– Retrieved $R_{eff}^a$ from in-situ measurements of around 0.18 ±0.02 μm, mean $\omega^a$ value of 0.96 as well as average $g_a$ exceeding 0.62 suggest a moderate absorbing properties of the plume and all properties place the BB plume in the lower part of the statistics performed by Nikonovas et al. (2015).

– Lidar profiles indicate the existence of BB plume at the level of 0 - 3.5 km with a complicated structure of sublayers limited by a number of (2- 5) temperature inversions. The presented issue also applies to the RH, which has a significant





impact on the $\omega^a$ profiles. The accuracy of modelled irradiances during the summer background conditions is considered to be sufficient deviating from the measured quantities by 2 % and 1 % considering $F_{in}$ and $F_{out}$ respectively while during the event the differences increase to 10 % and 5.8 % on average.

– Thus, we report modelled mean $RF_{surf}$, $RF_{atm}$ and $RF_{toa}$ at the level of -78.9 $\mathrm{Wm^{-2}}$, -47.0 $\mathrm{Wm^{-2}}$ and 31.9 $\mathrm{Wm^{-2}}$,

which indicates cooling effects at the surface and TOA while it reveals strong heating within the atmosphere. It might be translated into up to 2 $\mathrm{Kday^{-1}}$ of the $r_h$. Obtained values are consistent with results reported for the similar period and likely the same solar zenith angles performed by Stone et al. (2008). The comparison of modelled and measured $RFs_{surf}$ indicates underestimation regarding the first quantity, however, the latter should be interpreted with caution since it might be cloud contaminated.

– Averaged $RFE_{surf}$ reaches in this study -125.9 $\mathrm{Wm^{-2}}/\tau_{550}$ indicating higher values in comparison to $RFEs_{surf}$ obtained for wildfires from boreal regions, while for other fire sources it is considerably lower by 12 - 77 %. Authors believe the main reason, among different aerosol intensive properties, is distinct solar zenith angle and high value of daily mean solar radiation at TOA during Arctic summer.

– The discrepancies between modelled RFs and RFEs obtained for MODTRAN and robust Fu-Liou simulations oscillate

around 15 % with lower values usually attributed to the latter (exception for the atmospheric values). Considering different inputs and spatial resolution used for both simulations, the results are satisfactory. Also, the mean bias of RFs using the Independent Pixel Approximation approach in the vicinity of Ny-Alesund is estimated at the level of 2 $\mathrm{Wm^{-2}}$. Therefore, it almost reaches the magnitude of translating $\omega^d$ value into $\omega^a$ profile considered in this study.

– ILES indicate that the main impact of the BB plume on the atmospheric dynamics is a graduate vertical expansion

and positive displacement of the BB layer characterised by neutral stratification. The TKE in the simulated BB layer is around 0.3 $\mathrm{m^2 s^{-2}}$. In a reference simulation without effects from the BB plume included, the flow remained nearly non-turbulent throughout the simulation period.

Presented long-range transport of wildfires from Alaska to European Arctic indeed has a significant impact both on the radiative properties and atmospheric dynamics. We believe that the detailed studies on this topic are needed especially considering a

significant positive trend in mid-latitudes fires frequency during the summer season in the last 25 years and therefore a possible more frequent advection over Arctic region (Young et al., 2017). Thus, it is expected to have an impact on the annual Arctic RF means and Arctic amplification.

*Competing interests.* The authors declare no conflict of interests.



*Acknowledgements.* The authors would like to acknowledge the support of this research from the Polish-Norwegian Research Programme operated by the National Centre for Research and Development under the Norwegian Financial Mechanism 2009-2014 within the frame of Project Contract No Pol-Nor/196911/38/2013.

The authors are grateful for support from Marion Matturilli for providing data from the Baseline Surface Radiation Network (BSRN)
5   measured at AWIPEV station in Ny-Alesund.





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
