# Peer review of "Radiative Impact of an Extreme Arctic Biomass-Burning Event"

_Atmospheric Chemistry and Physics, 2017_

## Referee Comment (RC1) · Anonymous Referee #1 · 12 Feb 2018

The paper deals with radiative impact of biomass burning plume reaching to Svalbard, Arctic. It is very interesting topic and important for radiation budget and climate in the Arctic. However, the presentation of the result is so limited that sometimes it is difficult to follow exactly. In the manuscript, large part of the results are devoted to the comparison of the radiation code between MODTRAN and Fu-Liou (Fig. 4 and 5), and not so much description was made for the comparison with actual observed radiative fluxes. For example, Fig. 3 should be one of the main result to be shown; however, it is of some poor expression. In the figure caption, no explanation was made for observed flux (Rad F) and RF (Rad RF). I could not find any curves for Fu-Liou in the figure! If you stick to this comparison with large weights, then it be better to change your title. Also, why observed flux or RF has large gaps? The major aim of the paper is only

radiative effect, but that of BB plume. As for BB plume, we can only know very limited information from Fig. 2 (vertical distribution of extinction coefficients). I know that your group (including yourself as co-author) has already several papers related to this same BB and Markowicz et al. (2016a) shows comprehensive feature of BB plume. Even duplicated, some information be helpful to be shown in this paper also (for example, just like Fig. 2, 3, 4 or 10 in Markovicz et al., 2016a). (Specific comments) - Ny Alesund should be written "Ny-Ålesund". - Fig. 1: Though the figure occupies whole page, the information it shows seems to be not so interesting for the reader. Also, what is "white-sky albedo"? - P14, L5, 15, P16, L6: Relations with clouds are explained in several parts; however, we have no information on clouds in any figures. It is difficult to follow. - P15, L32, 34: What is Fcin or Fcout? There are no such symbols in Fig. 3. - Fig. 3: Explanation/ figure caption of Fig. 3 is limited. What is the large gaps in observed radiative fluxes (P16, L7 says radiometer data are removed – not easy to understand). There are no flux or RF of "within the atmosphere (subscript atm)" in the figure! There is no results by Fu-Liou. What is "Rad"? There is no explanation in the caption. We would like to know the data of $\tau$ (tau) itself. - P17, L17: RFE appears first, but no explanation here (only shown afterwards in P19, L16). - P20, L6: I have never heard of "Ny-Ålesund valley". Normally it is said as Ny-Ålesund fjord. - P23, L5: What is "LESs"? - Fig. 7: Is the wavy pattern in (a) meaningful? It seems to be rather artificial due to small change of vertical gradient of $\theta$ (T). - Conclusion: Items of conclusion seems to be different from results and discussions. To indicate these conclusions, you need to add more discussions to connect to these conclusions. - P25, L 8: What is "the first" and "the latter"? - P25, L10-11: RFEssurf obtained for wild fires from boreal regions, -→ any reference? - P25. L19: What is "ILES"? - P25, L24: Impact on the atmospheric dynamics is not clearly described in the manuscript. - P25, L26-27: The meaning of the sentence "Thus, it is expected. . ." is not clear. - References: Descriptions are not complete in some, for example, Markowicz et al., 2002, or — 2017b, Stone et al., 2008, Wang et al., 2006.

---

## Author Comment (AC1) · 7 Mar 2018

*Italic font style denotes the Referee comments,* while normal font - our answer.

**General comments**

*The paper deals with radiative impact of biomass burning plume reaching to Svalbard, Arctic. It is very interesting topic and important for radiation budget and climate in the Arctic. However, the presentation of the result is so limited that sometimes it is difficult to follow exactly.*

Thank you for your prompt and kind review also for specifying issues and points that we can improve. We hope that the changes we proposed, listed below, shall satisfy the reviewer.

*In the manuscript, large part of the results are devoted to the comparison of the radiation code between MODTRAN and Fu-Liou (Fig. 4 and 5)*

In the revised version the chapter "3.3 The comparison of RF derived from MODTRAN and Fu-Liou simulations" is shortened - only a main outcome is left for this section, namely a brief information on the performance of our custom code to a robust model, as this is not the main result we wanted to emphasize. We moved both figures (Fig. 4 and 5) to the appendix.

*not so much description was made for the comparison with actual observed radiative fluxes.*

In the revised manuscript, this comparison is added together with the according figure.

*For example, Fig. 3 should be one of the main result to be shown; however, it is of some poor expression. In the figure caption, no explanation was made for observed flux (Rad F) and RF (Rad RF). I could not find any curves for Fu-Liou in the figure!*

Indeed, the Referee is right. We missed that the caption is ill-copied and should be as following:

Temporal variability of (a) radiation fluxes: total incoming flux with the presence of aerosols (black) and without aerosol load (blue) as well as total outgoing flux (red) at the surface simulated (dots) by MODTAN and measured by radiometers (lines). Sub-figure (b) presents radiative forcing at the surface (black) and at the top of atmosphere (red).

*Also, why observed flux or RF has large gaps?*

The explanation is included on P15 L12-L16, quoted below: *Figure 3 presents the comparison of irradiances (Fig. 3a) and clear-sky RF (Fig. 3b) obtained by the means of MODTRAN simulations and estimated both by the radiometers' measurements in Ny-Alesund and model calculations (reference case) for the BB2015 event. The latter represent all-sky conditions since the discussed BB event is extremely complicated and therefore a possible cloud contamination seems to be impossible to separate entirely. However, periods with a clear influence of clouds 15 were removed, therefore presented the mean value of RF lacks most intense period (see Fig. 3b).*

We added a short note in the figure caption to emphasize the above information. As this chapter is one of the main results in the paper, we will work on its better expression as now we realized that not everything is clear to the reader.

*The major aim of the paper is only radiative effect, but that of BB plume. As for BB plume, we can only know very limited information from Fig. 2 (vertical distribution of extinction coefficients). I know that your group (including yourself as co-author) has already several papers related to this same BB and Markowicz et al. (2016a) shows comprehensive feature of BB plume. Even duplicated, some information be helpful to be shown in this paper also (for example, just like Fig. 2, 3, 4 or 10 in Markowicz et al., 2016a).*

Thank you for this comment. We added a new (brief) chapter about the overall characteristics of the event in terms of aerosol optical properties. It is based on the similar figure to Fig. 10 from Markowicz et al., 2016a, highlighting the temporal variability of extinction coefficient profiles and AOD. Please see the appendix for this comment. (hint: black lines refer to cloud occurrence retrieved from observations).

**Specific comments**

In the revised version all of the specific comments were followed and corrected. Thank you very much for careful revision.

[Figure]

**Fig. 1.** Aerosol optical properties during the biomass-burning event retrieved from NAAPS model and from observations.

---

## Referee Comment (RC2) · Anonymous Referee #2 · 20 Mar 2018

**Review of the manuscript "Impact of a Strong Biomass Burning Event on the Radiative Forcing in the Arctic" by Justyna Lisok et al.**

**General comment:**

This study investigates the radiative impacts in the Arctic (Ny-Ålesund, Svalbard) of biomass burning aerosols transported from Alaska during an especially intense event. Authors combine in-situ and remote sensing measurements, radiative transfer models, as well as Lagrangian, Eulerian and LES models to investigate the consequences of this event. This work is scientifically significant and of good quality, and is thus suitable for ACP. I have one major comment concerning the presentation of the results; I had difficulties understanding several sections of the manuscript, especially the introduction, due to shortcomings in the English language used. Details are given in the specific comments below.

**Major comment:**

1. I found it difficult to fully assess the quality of the paper due to English language and grammar issues, which should be addressed before publication. Due to the number of such errors I am not able to point them all out in this review. As an example, it can be hard to understand the following sentences:

"The effect of BB aerosol from the regional point of view is claimed to have stronger temporal variations indicating the change of the regional climate patterns (Wang et al., 2006) It might be especially important over the bright surfaces regarding changes in the surface and cloud albedo (Screen and Simmonds, 2010), which in particular may indicate a positive $RF_{toa}$"

**Specific comments:**

2. Along with Myhre et al. (2013), the manuscript could include a citation to Sand et al., (2017), who investigated specifically the radiative forcing of aerosols in the Arctic in the AeroCom phase II models.
3. P2., l. 16, For IPCC results, Myhre et al. (2013b) might be a better reference than Pachauri et al., (2014)
4. P.2, l.34: Do you really mean reducing "values", not reducing data coverage?
5. P.4, ll. 2-10: I think authors should clearly indicate here their own new/original contributions in this paper, and what work (e.g. simulations) was already performed for previous studies such as Markowicz et al. (2017b).
6. P7., section 2.2: If I understand correctly, the in-situ measurements (e.g. SMPS, PSAP), are performed at the surface. Can you give reasons why these values are representative of the whole column, since the plumes extends at relatively high altitudes, and the Arctic surface and free troposphere are often decoupled.
7. P.7, l. 12: The a and d superscripts should be explained there, when they are first introduced, and not on page 8.
8. P. 9, equations 7 and 8: The text mentions $RF_{net}$ and $RF_{rel}$, but the equations give $F_{net}$ and $f_{rel}$.
9. P.9, l.15: If this product is from MODIS, this should be indicated.
10. P.9 l. 22: The "BRDF" acronym should be explained here.
11. P. 12 ll. 1-5: You could also compare to single scattering albedos used by Lund Myhre et al. (2007).
12. P. 12 l. 20: Is PM10 really reported in ppb, not µg m-3?
13. P. 15 l. 10 "and additional no change in the irradiances from the reference simulation" it is not clear what you mean by this sentence.
14. P. 15, l. 16: what do you mean here by a "real" value of albedo?

15. Figure 3. There are several issues with this figure. First, the caption does not seem to match the contents, as the "Rad" quantities, which seem to be observations, are not explained in the caption. The caption mentions Fu-Liou results that are apparently not shown. The quantities do not seem to be daily mean values. In addition, RF quantities in panel b should use different colors/symbols than the F results in panel a, as the current choices is very confusing.

16. Figure 3: What are the reasons for the differences between F and ModF results at the end of the period, after 12h on 11 July?

17. Pp. 15, 16: This section should include more paragraphs breaks to better separate the different ideas.

18. P. 16, l. 6: How would increase turbulence lead to higher variability in F_in?

19. P. 17, l. 17: Explain the meaning of "RFE" when it is first introduced. For what reason is RFE a more accurate quantity for intercomparisons?

20. P. 17, l. 31: It is not clear here for someone unfamiliar with these codes that DISORT is included within MODTRAN and not a standalone radiative transfer model. Consider rephrasing this sentence.

21. P. 17, l. 31 and elsewhere: Can you explain what you mean by "robust" when referring to Fu-Liou? Do you mean more detailed?

22. Pp. 18-19: This section should include more paragraphs breaks to better separate the different ideas.

23. P. 18, ll. 13-18: I do not think it is needed here to remind the meaning of the different colors on Figure 4, since they are already explained on the Figure.

24. P. 19, l.4 and elsewhere: The correct reference is Lund Myhre et al. (2007), not Myhre et al., since "Lund Myhre" is the last name of the first author.

25. P. 19, ll. 12-15: This section would be clearer if the analysis of Figure 4 started with this remark, since the most obvious result from Figure 4 is that there is a very good agreement for RF between MODTRAN and Fu-Liou.

26. Figure 5: What are the reasons for the strong differences in RFE between MODTRAN and Fu-Liou for 9 July?

27. P. 20, l.7: "In the previous sections, we discussed the RF computed for a single cell" maybe this should also be mentioned explicitly in the beginning of the previous sections, e.g. at the beginning of 3.2.

28. P. 20, ll. 13-14: Why not show RF directly, instead of this relative value? This should maybe be explained when the equations are discussed.

29. Figure 6: There are also several issues with this figure. First, the colorbar should include a label. Since values go from negative to positive, it would be a lot clearer to use a divergence colormap where 0 is indicated by a special color, for example white. It is also unclear to a reader unfamiliar with the "ICA" terminology what is the exact difference between panels a and b. I understand that the point is to study the effect of e.g. topography on the RF calculations, but consider writing a more explicit caption, and consider including in the text an explanation of the difference between these two calculations and the aim of this 2-panel comparison.

30. Figure 6: Results seem to show a negative RF over high-albedo surfaces. Other studies (e.g. Sand et al., 2017) often showed a positive RF of BB aerosols over snow and ice. Is this due to the high single-scattering albedo here? To a relatively low surface albedo compared to typical snow and ice-covered surfaces in the Arctic?

31. Conclusion: If possible, use the full name of the quantities discussed in the conclusion, e.g. "heating rate", instead of the "rh" notation.

32. P. 25, l. 4: Are these average values? Over what time window?

33. P. 25, l. 7: Are you really comparing modelled RF to observations in this study?

**ADDITIONAL REFERENCES:**

Sand, M., Samset, B. H., Balkanski, Y., Bauer, S., Bellouin, N., Berntsen, T. K., Bian, H., Chin, M., Diehl, T., Easter, R., Ghan, S. J., Iversen, T., Kirkevåg, A., Lamarque, J.-F., Lin, G., Liu, X., Luo, G., Myhre, G., Noije, T. V., Penner, J. E., Schulz, M., Seland, Ø., Skeie, R. B., Stier, P., Takemura, T., Tsigaridis, K., Yu, F., Zhang, K., and Zhang, H.: Aerosols at the poles: an AeroCom Phase II multi-model evaluation, Atmos. Chem. Phys., 17, 12197-12218, https://doi.org/10.5194/acp-17-12197-2017, 2017.

Myhre, G., D. Shindell, F.-M. Bréon, W. Collins, J. Fuglestvedt, J. Huang, D. Koch, J.-F. Lamarque, D. Lee, B. Mendoza, T. Nakajima, A. Robock, G. Stephens, T. Takemura, and H. Zhang, 2013: Anthropogenic and natural radiative forcing. In *Climate Change 2013: The Physical Science Basis. Contribution of Working Group I to the Fifth Assessment Report of the Intergovernmental Panel on Climate Change*. T.F. Stocker, D. Qin, G.-K. Plattner, M. Tignor, S.K. Allen, J. Doschung, A. Nauels, Y. Xia, V. Bex, and P.M. Midgley, Eds. Cambridge University Press, pp. 659-740, doi:10.1017/CBO9781107415324.018.

---

## Author Response (AR1)

**Authors' Response to the Referees' Comments on "Impact of a Strong Biomass Burning Event on the Radiative Forcing in the Arctic" Lisok et al.**

April 17, 2018

*Italic font style denotes the Referee comments,* while normal font - our answer.

**The Anonymous Referee #1**

**General comments**

*The paper deals with radiative impact of biomass burning plume reaching to Svalbard, Arctic. It is very interesting topic and important for radiation budget and climate in the Arctic. However, the presentation of the result is so limited that sometimes it is difficult to follow exactly.*

Thank you for your prompt and kind review also for specifying issues and points that we can improve. We hope that the changes we proposed, listed below, shall satisfy the reviewer.

*In the manuscript, large part of the results are devoted to the comparison of the radiation code between MODTRAN and Fu-Liou (Fig. 4 and 5), and not so much description was made for the comparison with actual observed radiative fluxes.*

In the revised version the chapter "3.3 The comparison of RF derived from MODTRAN and Fu-Liou simulations" is shortened - only a main outcome is left for this section, namely a brief information on the performance of our custom code to a fast model, as this is not the main result we wanted to emphasize. We moved the figure (Fig. 5) to the appendix.

Additionally, we added a new section concerning the comparison between modeled and measured irradiances (see section 3.3 of the revised manuscript).

*For example, Fig. 3 should be one of the main result to be shown; however, it is of some poor expression. In the figure caption, no explanation was made for observed flux (Rad F) and RF (Rad RF). I could not find any curves for Fu-Liou in the figure!*

Indeed, the Referee is right. We missed that the caption is ill-copied and should be as following:

Temporal variability of (a) the surface radiation fluxes: total incoming flux with the presence of aerosols $F_{in}$ (black) and without aerosol load $F_{cin}$ (blue), as well as total outgoing flux $F_{out}$ (red), simulated by MODTAN (dots) and measured by radiometers (lines). The gaps in the radiometer data refer to the cloud contamination. Sub-figure (b) presents radiative forcing at the surface $RF_{surf}$ (green) and at the top of atmosphere $RF_{toa}$ (orange).

*Also, why observed flux or RF has large gaps?*

The explanation is included on P15 L30-L35, quoted below: *Radiometer data represent all-sky conditions, since the discussed BB event is extremely complicated and therefore a possible cloud contamination seems to be impossible to separate entirely. However, periods with a clear influence of clouds were removed (i.e. 15:00-21:00 $10^{th}$ July), therefore the presented mean value of RF, lacks the most intense period (see Fig. 4b).*

We added a short note in the figure caption to emphasize the above information. As this chapter is one of the main results in the paper, we rephrased it to state our result more clearly.

*The major aim of the paper is only radiative effect, but that of BB plume. As for BB plume, we can only know very limited information from Fig. 2 (vertical distribution of extinction coefficients). I know that your group (including yourself as co-author) has already several papers related to this same BB and Markowicz et al. (2016a) shows comprehensive feature of BB plume. Even duplicated, some information be helpful to be shown in this paper also (for example, just like Fig. 2, 3, 4 or 10 in Markowicz et al., 2016a).*

Thank you for this comment. We added a new (brief) chapter about the overall characteristics of the event in terms of aerosol optical properties. It is based on the similar figure to Fig. 10 from Markowicz et al., 2016a, highlighting the temporal variability of the optical and microphysical properties of the aerosol.

**Specific comments**

*Ny-Alesund should be written "Ny-Ålesund"*

Corrected as suggested.

*Fig. 1: Though the figure occupies whole page, the information it shows seems to be not so interesting for the reader. Also, what is "white-sky albedo"?*

The figure was moved to the appendix and the meaning of white-sky albedo was explained. This is the method (also referred to as bihemispherical albedo), that applies the integration of BRDF over all viewing directions using only the diffuse irradiance.

*P14, L5, 15, P16, L6: Relations with clouds are explained in several parts; however, we have no information on clouds in any figures. It is difficult to follow.*

In the revised manuscript, Fig. 1b presents the occurrence of clouds in the respective period of time.

*P15, L32, 34: What is Fcin or Fcout? There are no such symbols in Fig. 3. Fig. 3: Explanation/ figure caption of Fig. 3 is limited. What is the large gaps in observed radiative fluxes (P16, L7 says radiometer data are removed – not easy to understand). There are no flux or RF of "within the atmosphere (subscript atm)" in the figure! There is no results by Fu-Liou. What is "Rad"? There is no explanation in the caption.*

Fcin or Fcout refer to the respective (total incoming and outgoing) fluxes calculated under the reference simulation (no aerosol load). As previously stated, the figure caption was ill-copied. In the revised manuscript this error was corrected. Large gaps in the radiometer data are indicated by the occurance of the clouds, we added additional information in the figure caption.

*We would like to know the data of $\tau$ itself.*

The information of the temporal variability of optical and microphysical properties (together with with $\tau$) were added in the Fig. 1 along with the according explanatory section.

*P17, L17: RFE appears first, but no explanation here (only shown afterwards in P19, L16).*

Corrected as suggested.

*P20, L6: I have never heard of "Ny-Ålesund valley". Normally it is said as Ny-Ålesund fjord.*

The referee is right. We used the Norwegian name (Kongsfjorden) instead.

*P23, L5: What is "LESs"?*

We apologize for the typo. Nevertheless, the explanation of ILES is included when the EULAG model is described (section 2.1). In the line under consideration, we added the according note in the brackets "(see section 2.1)" after the abbreviation.

*Fig. 7: Is the wavy pattern in (a) meaningful? It seems to be rather artificial due to small change of vertical gradient of (T).*

The vertical variability of a heating rate at a reference simulation (fig. 7a) used for EULAG calculations is vastly connected with the shape of specific humidity profile, and accordingly with the water vapor absorption bands. The heating rate at the so called polluted simulation (fig. 7d) additionally is a function of both the single-scattering albedo and aerosol load in the layers (the extinction coefficient profile).

*Conclusion: Items of conclusion seems to be different from results and discussions. To indicate these conclusions, you need to add more discussions to connect to these conclusions.*

We kindly disagree with the referee. The sentences used in the conclusions are almost a literal copy of the statements from the results. We may speculate that the fist version of the manuscript was a bit chaotic and therefore the main outcome from specific sections may have been missed. We hope, that in the revised manuscript, this issue doesn't appear any more.

*P25, L 8: What is "the first" and "the latter".*

Thank you, we rephrased the sentence.

*P25, L10-11: RFEssurf obtained for wild fires from boreal regions, - any reference?*

Corrected by adding Markowicz et al., 2016b, Markowicz et al., 2002 and additionally Garcia et al., 2012. Thank you.

*P25. L19: What is "ILES"?*

The explanation of ILES (Implicit Large Eddy Simulations) is included when the EULAG model is described (section 2.1). We added the reference note in the section 3.7 when it is mentioned for the first time in the results.

*P25, L24: Impact on the atmospheric dynamics is not clearly described in the manuscript. - P25, L26-27: The meaning of the sentence "Thus, it is expected ... " is not clear.*

Both sentences revealed shortcomings in English, we apologize for that. In the revised manuscript they were rephrased as following:

*In this study we have shown that long-range transport of wildfire aerosols from Alaska to European Arctic, certainly has a significant impact on radiative properties. Furthermore, our results also indicate an impact on atmospheric dynamics. We believe that the detailed studies on this topic are needed, especially considering a significant positive trend in mid-latitudes fire frequency during the summer season in the last 25 years; and therefore possibly more frequent advection over the Arctic region (Young et al., 2016)*

*References: Descriptions are not complete in some, for example, Markowicz et al., 2002, or — 2017b, Stone et al., 2008, Wang et al., 2006*

Thank you, we improved this section.

**The Anonymous Referee #2**

We wanted to thank the reviewer for raising issues that limit the understanding of the paper as it helped us to improve the paper. We hope that the reviewer will be satisfied with the changes made to the new version of the paper.

**Major comment**

*[...] I found it difficult to fully assess the quality of the paper due to English language and grammar issues, which should be addressed before publication. Due to the number of such errors I am not able to point them all out in this review. [...]*

The paper went through a major reorganization regarding English shortcomings along with English-proof reading.

**Specific comments**

*Along with Myhre et al. (2013), the manuscript could include a citation to Sand et al., (2017), who investigated specifically the radiative forcing of aerosols in the Arctic in the AeroCom phase II models.*

Indeed, both papers were significant for the section, thank you.

*P2., l. 16, For IPCC results, Myhre et al. (2013b) might be a better reference than Pachauri et al., (2014)*

The referee is right. Corrected.

*P.2, l.34: Do you really mean reducing "values", not reducing data coverage?*

True, we meant 'data coverage', thank you.

*P.4, ll. 2 - 10: I think authors should clearly indicate here their own new/original contributions in this paper , and what work (e.g. simulations) was already performed for previous studies such as Markowicz et al. (2017b).*

Corrected as requested. In the revised paper the section is written as follows:

*Previously presented by scientific papers, and characterized in this research, was the study of smoke transport over the Arctic during July 2015. Markowicz et al., 2016a reported the temporal and spatial variability of aerosol single-scattering properties measured by in situ and ground-based remote sensing instruments over Svalbard and in Andenes, Norway. Moroni et al., 2017, discussed morphochemical characteristics and mixing state of smoke particles in Ny-Ålesund as indicated by DEKATI 12-stage low volume impactor, combined with scanning electron microscopy. Markowicz et al., 2017b on the other hand, presented a comprehensive description of smoke radiative and optical properties on a regional scale. The paper examined ageing processes of the smoke plume under study, while transported from the source region across the High Arctic. Simple Fu-Liou radiative transfer model, combined with NAAPS aerosol transport model, were used to determine the spatial distribution of aerosol single-scattering properties and RFs for the period of 5-15 July 2015, in the area to the north of 55ºN, where the transport of BB aerosol was observed.*

*In this paper, we utilise MODTRAN radiative transfer simulations and aerosol optical properties obtained from in situ and ground-based remote sensing instruments, to retrieve clear-sky direct RF over the area close to Ny-Ålesund. The research aims to estimate the biases connected with (i) hygroscopicity, (ii) variability of $\omega$ profiles, and (iii) plane-parallel closure of the modeled atmosphere. The main outcome of this research is the implementation of new methodology to retrieve the profile of $\omega$ at ambient conditions, utilising in situ measurements and lidar profiles (section 3.2). Simulated RFs were compared to simple radiative transfer model (section 3.5). Section 3.6 shows an example of RF distribution at the surface, in the vicinity of Kongsfjorden. The last part presents the influence of unstably stratified biomass burning air masses on the turbulence development, which is shown in section 3.7. Additionally, we confirmed the source region of the BB plume. A chemical weather model with satellite-derived biomass burning emissions was used to interpret the transport and transformations pathways.*

*P7., section 2.2 : If I understand correctly, the in - situ measurements (e.g. SMPS, PSAP), are performed at the surface. Can you give reasons why these values are representative of the*

*whole column , since the plumes extends at relatively high altitudes, and the Arctic surface and free troposphere are often decoupled.*

In the revised manuscript, we added an explanatory section concerning our assumptions to $\omega$ and $g$ retrieval, quoted below:

*Vertical profiles of single-scattering properties at ambient conditions are used as input parameters to MODTRAN and Monte Carlo calculations. The retrieval is based on the in situ single-scattering properties, measured at the surface in dry conditions (denoted later on as superscript 'd'), and on vertical profiles of $\sigma_{ext}^{a}$, as well as RH at ambient conditions (hereinafter superscript 'a') from KARL lidar and radio-sounding data.*

*In the reference to temporal variability of range-corrected signal, measured at 532 nm by Micropulse Lidar, Markowicz et al, 2016a, characterize smoke plume as a rather well-mixed layer of BB aerosol extending from around 4 - 6 km on $9^{th}$ to 0 - 3.5 km later on. Both contributions of BB-like aerosol in the NAAPS AOD, estimated on the level as high as 80%, and the similarity between columnar and in situ aerosol extensive properties such as $\alpha$ (Markowicz et al, 2016a), suggest that smoke plume may have crossed PBL and mixed with the lowermost part of the troposphere. Additionally, very little aerosol load existing above smoke plume plays a minor role in affecting the radiative properties of the atmosphere and therefore may be neglected. This is why, in the presented methodology, we assume no changes in chemical composition vertically, so that most of the possible vertical variability of $\omega^{a}$ at ambient conditions, is attributed to changes in RH. Therefore, we approximate initial profiles of $\omega^{d}$ and $R_{eff}^{d}$ by setting them up to the values of in situ measurements and consider them constant with altitude. By introducing hygroscopic growth model for particles with known size distribution, one may obtain $\omega^{a}$ profile as well as $g^{a}$.*

*P.7, l. 12: The a and d superscripts should be explained there, when they are first introduced, and not on page 8.*

Corrected as suggested.

*P. 9, equations 7 and 8: The text mentions RFnet and RFrel, but the equations give Fnet and frel.*

Thank you, this was our mistake while copy-pasting to latex.

*P.9, l.15 : If this product is from MODIS, this should be indicated.*

Indeed, thank you.

*P.9 l. 22: The "BRDF" acronym should be explained here.*

Corrected.

*P. 12 ll. 1 - 5: You could also compare to single scattering albedos used by Lund Myhre et al. (2007).*

Thank you for your helpful comment, we referred also to Lund Myhre et al. (2007) in the section under consideration.

*P. 12 l. 20: Is PM10 really reported in ppb, not $\mu gm^{-3}$ ?*

Thank you, the text was corrected to "the mass mixing ratio".

*P. 15 l. 10 " and additional no change in the irradiances from the reference simulation" it is not clear what you mean by this sentence.*

Indeed, we rephrased the sentence.

*P. 15, l. 16: what do you mean here by a "real" value of albedo?*

Indeed, we rephrased the sentence.

*Figure 3. There are several issues with this figure. First, the caption does not seem to match the contents, as the "Rad" quantities, which seem to be observations, are not explained in the caption. The caption mentions Fu - Liou results that are apparently not shown. The quantities do not seem to be daily mean values. In addition, RF quantities in panel b should use different colors/symbols than the F results in panel a , as the current choices is very confusing.*

Thank you, we didn't notice that the caption was ill-copied. In the revised manuscript, the caption matches the figure.
We changed the colors/symbols in the b panel for the clarity.

*Figure 3: What are the reasons for the differences between F and ModF results at the end of the period, after 12h on 11 July ?*

This difference is a result of low cloud appearance at around noon $11^{th}$ July, as explained in the section 3.1. In the revised version of the manuscript we removed all cloud-contaminated data from this figure, also the $F_{in}$ after 11:30 July 11.

*Pp. 15, 16: This section should include more paragraphs breaks to better separate the different ideas.*

The paragraph breaks were added.

*P. 16, l. 6: How would increase turbulence lead to higher variability in $F_{in}$?*

We apologize for this linguistic shortcoming. The higher variability of $F_{in}$ on $10^{th}$ is a direct effect of the appearance of cumulus clouds. They, in turn, result from: (1) the aerosol activation based on the most common mechanism of cloud formation and (2) the instability of the atmospheric dynamics, as this is the reason why cumulus clouds are formed rather than other clouds.
After rephrasing, this sentence should be as follows:

*We may expect that higher variability of Rad $F_{in}$, visible by comparison to the $9^{th}$ July, together with an appearance of clouds inside the smoke plume, are likely to result from both a possible BB aerosol activation and increased turbulence. Further to this, a number of high- and mid-level cumulus clouds are reported around noon and in the afternoon (Markowicz et al., 2016).*

*P. 17, l. 17: Explain the meaning of "RFE" when it is first introduced. For what reason is RFE a more accurate quantity for intercomparisons?*

Corrected as suggested. RFE is a more accurate quantity for inter-comparison only when intrinsic properties of the plume are taken under consideration as it was stated in the further part of the sentence. However, in the revised version of the manuscript this sentence, after rephrasing of the paragraph was omitted.

*P. 17, l. 31: It is not clear here for someone unfamiliar with these codes that DISORT is included with in MODTRAN and not a standalone radiative transfer model. Consider rephrasing this sentence.*

Corrected as suggested.

*P. 17, l. 31 and elsewhere: Can you explain what you mean by "robust" when referring to Fu - Liou? Do you mean more detailed?*

We apologize for this ill-translation. We meant 'fast' and 'less-complicated' in terms of solvers of the radiative transfer equations. It was improved in the revised manuscript.

*Pp. 18 - 19: This section should include more paragraphs breaks to better separate the different ideas.*

Corrected as suggested.

*P. 18, ll. 13 - 18: I do not think it is needed here to remind the meaning of the different colors on Figure 4, since they are already explained on the Figure.*

We agree, thank you for this suggestion.

*P. 19, l. 4 and elsewhere: The correct reference is Lund Myhre et al. (2007), not Myhre et al., since "Lund Myhre" is the last name of the first author.*

Corrected as suggested.

*P. 19, l. 12 - 15 : This section would be clearer if the analysis of Figure 4 started with this remark , since the most obvious result from Figure 4 is that there is a very good agreement for RF between MODTRAN and Fu - Liou.*

We agree with the reviewer and changed the text accordingly.

*Figure 5: What are the reasons for the strong differences in RFE between MODTRAN and Fu - Liou for 9 July ?*

The main reason for the modeled discrepancies in $RFE$ are (1) the differences in inputs to models, in particular the assumed aerosol optical properties and secondarily PW as well as (2) the distinction between solvers of the radiative transfer equations used in both models, that may give different results even though the exact inputs are assumed. The latter issue is more widely described in the following paper: Myhre, G. et al,2009:Intercomparison of radiative forcing calculations of stratospheric water vapour and contrails, METEOROL Z, 18(6), pp585-596.

Note, this part of the section was moved to the appendix B. This was requested by the Referee 1 being concerned that the inter-comparison between RTM models was not the main subject of the manuscript and additionally unnecessarily lengthened the paper.

*P. 20, l. 7: " In the previous sections, we discussed the RF computed for a single cell " maybe this should also be mentioned explicitly in the beginning of the previous sections, e.g. at the beginning of 3.2.*

We decided to add this information in the description of models.

*P. 20, l l . 13 - 14: Why not show RF directly, instead of this relative value? This should maybe be explained when the equations are discussed.*

In the revised manuscript, we added the following information to the 2.3.4 section with 3D Monte Carlo equations:

*The results from 3D Monte Carlo model, as mentioned earlier, are used to characterise spatial variability of RF and therefore to diagnose possible uncertainties resulting from using single-column radiative transfer models, represented by MODTRAN and Fu-Liou codes. Taking into account the above goals, we resigned from performing time-consuming simulations of daily mean broadband RFs for the model domain; and instead we relied on the relative value of RF calculated for 1 $\lambda$, with respect to its value at TOA at a given zenith angle. Such an approach allowed for defining higher spatial resolution.*

*Figure 6: There are also several issues with this figure. First, the colorbar should include a label. Since values go from negative to positive, it would be a lot clearer to use a divergence colormap where 0 is indicated by a special color, for example white. It is also unclear to a reader unfamiliar with the "ICA" terminology what is the exact difference between panels a and b. I understand that the point is to study the effect of e.g. topog raphy on the RF calculations, but consider writing a more explicit caption, and consider including in the text an explanation of the difference between these two calculations and the aim of this 2 - panel comparison.*

The label to the colorbar was added. Regarding the divergence colormap, we kindly disagree with the referee, as this would limit the number of colors used for the negative RFs. As the area of a positive RF is very small, we feel that this change is not of a great importance. Instead we added a black line to the colorbar highlighting 0 value.

*Figure 6: Results seem to show a negative RF over high - albedo surfaces. Other studies (e.g. Sand et al., 2017) often showed a positive RF of BB aerosols over snow and ice. Is this due to the high single - scattering albedo here ? To a relatively low surface albedo c ompared to typical snow and ice - co vered surfaces in the Arctic ?*

Fig.6 in the manuscript and the work by Sand et al. (2017) present radiative forcing at different levels. The figure shows aerosol radiative forcing at the surface while Sand et al (2017) at the top of the atmosphere. Aerosol radiative forcing at the surface is typically negative.

*Conclusion: If possible, use the full name of the quantities discussed in the conclusion , e.g. "heating rate", instead of the "rh" notation .*

Corrected as suggested.

*P. 25, l. 4: Are these average values? Over what time window?*

This averages refer to the BB event, in particular 14:00 July $9^{th}$ - 11:30 July $11^{th}$. We changed the sentence accordingly.

*P. 25, l . 7: Are you really comparing modelled RF to observations in this study?*

We apologize for this shortcoming in English. We have changed the sentence to match the actual meaning. Nevertheless, note that we also added a comparison of modeled and measured Fs.

[revised manuscript text omitted]

---

## Referee Report (RR1)

**Review of "Impact of a Strong Biomass Burning Event on the Radiative Forcing in the Arctic" by Lisok et al.**

**General comment:**

This study investigating the intense radiative impacts in the Arctic (Ny-Ålesund, Svalbard) of an extreme biomass burning event is scientifically significant and of good scientific quality, and is in my opinion appropriate for ACP. However, I think there are serious issues with the English language. Specifically, I found that, while some sections (e.g. 2.2, 2.3, 3.7) are well written, most sections of the manuscript (and especially the introduction and Sections 3.4) are very difficult to understand even by someone already quite familiar with the topic. As a result, I think the paper fails to pass these ACP criteria

(https://www.atmospheric-chemistry-and-physics.net/peer_review/review_criteria.html)

10- Is the overall presentation well structured and clear?

11- Is the language fluent and precise?

I think the authors should have another look at the English language in the whole paper, and in the indicated sections in particular.

**Comments (line numbers and pages refer to the revised version of the manuscript with tracked changes):**

1.  Title: "impact ….on the radiative forcing in the Arctic", the wording is strange, in my opinion it should be instead "Radiative forcing of a strong BB event" or (better) "Radiative impact of a strong BB event"

2.  Abstract: "the factor of 10 in comparison to the average" should be "a factor of 10 above the average"

3.  P1 l17: "2.0 Pg of carbon aerosols is released into the atmosphere **by fires** each year", for now the text makes it seems like this includes all sources.

4.  **P2 L3: "90% of which are present in the fine mode (sizes XX nm – XX nm)"**

5.  P2 L5-18: This section is unfortunately extremely difficult to read, please rephrase.

6.  P2 L16: Please rephrase, BB plumes are darker than bright clouds or snow but do not directly decrease the albedo of clouds and snow (unless this sentence is about absorbing aerosols deposited on snow or cloud/aerosol interactions).

7.  P2 L24 This is also very unclear. "it is unlikely", what is "It" ?

8.  P2 L28 Please define AERONET and add a reference

9.  P3 L26, I think you should use "in the vicinity of Ny-Ålesund (Svalbard)" here instead of "Kongsfjorden", since "Kongsfjorden", will not be known by most readers. I understand that this was implemented to remove "Ny-Ålesund valley" from the manuscript, but in most cases it seems like using "Ny-Ålesund" or "near Ny-Ålesund" would be clearer than using "Kongsfjorden".

10. P3 L 27 A section number is missing.

11. P4 L9-10: If simulation names are introduced, they should be mentioned later in the text when appropriate, e.g. when showing simulation results in Figure 3, mention that this is the 'polluted' simulation.

12. P4 L7: I think you should mention there that thermodynamical variables are "from radiosonde measurements (see section ??)", for now it seems like you say that they are from HITRAN.

13. P4 L 11: Say why you also use Fu-Liou, if MODTRAN is the main RTM used in the study.

14. P5 L15: This should also say what GEM-AQ is used for.

15. P7 L22: What do you mean by "generalized"? Do you mean "regridded" or "upscaled"?

16. P9 L7: Does this apply to aged BB aerosols?

17. P10 L 23: For clarity I suggest "were changed **by a factor** of 10"

18. P11 fig1: Replace "tau/alpha" by "tau & alpha", for now it seems like the ratio of the 2 is plotted. Same for omega/CL.

19. P11 L6 "variability of alpha was rather stable" should be replaced by "variability of alpha was rather low" or "rather limited".

20. P12 L26 remove "may" in "may support the above statement"

21. P12 L30: Please rephrase, "exemplary" here is confusing, since this word usually means "perfect, exceptional" and also less commonly "providing an example".

22. P13 Fig2. There is a problem with the legend and labels in this figure. For example the green line is supposed to be sigma_a_ext but is in fact only sigma_a_ext observed by LIDAR. The legend should say specifically that green is observed sigma_ext, or lines in the second column (GEM-AQ) should also be in green. Columns should also be labeled to make this clearer, e.g. column 2 should be "GEM-AQ sigma_a_ext", column 1 should be "observed …"

23. P14 L1. It says there that in the last case vertical mixing is suppressed, when the following paragraph only mentions "the existence of vertical mixing". Where is this suppression shown or discussed?

24. P14 l6, replace "the next day" by "the 11$^{th}$", since just above the "10$^{th}$ and 11$^{th}$" are mentioned, which would mean that "the next day" is the 12$^{th}$.

25. P14 L17 Do you really mean "resulting from" (smoke mixing creates cumulus) or do you mean "resulting in" (cumulus lead to smoke mixing)?

26. P14 L20: Please rephrase the last sentence, which I could not understand as is.

27. Section 3.3: I think you should mention here that (if I'm not mistaken) part of this good correlation is due to the daily cycle in insolation.

28. P15 L 2 I think the following would be clearer: "**occasionally** represent all-sky conditions"

29. P15 L11 and elsewhere: Is "translation" the right term here? Maybe "correction"? This is also present later in the text.

30. P15 L13-15 There's a problem here with how the parentheses are placed, for now it seems like radiometer measurements are labeled as "polluted simulation".

31. P15 L20-24: this is also very confusing, consider separating into 2 sentences, one comparing model and measurements, and one comparing the 2 model simulations.

32. Section 3.4. The naming of the modeled quantities is confusing, since the simulations are called "polluted" and "reference", but the model results use the index "c" (e.g. $F\_c\_in$) for the "reference" simulation and no index for the polluted case. Maybe use $F\_pol\_in$ and $F\_ref\_in$ for clarity (or p and r). I suppose "c" stands for "clean" but you present this simulation as a "reference", not "clean" case. This is also not clear in the caption of Figure 4, where you say "no aerosol load" instead of "reference aerosols".

33. P16 L13-19. This section is extremely hard to understand, please rephrase and correct the English.

34. P 16 L 33-35. You mention 3 variables but then mention discrepancies between "both variables". What variables are these? Do you mean "all variables"?

35. P17 L2: There should be other / more general references (e.g. review papers, multimodel analysis, IPCC) than Stone et al. (2008) mentioning this, since this is a well-known feature of the RF of aerosols.

36. P17 L 10 "the RF sign **at TOA**"

37. P17 L11-16: If I'm not mistaken, if both BC and OC increase a lot and sulfate increases less, than the ratio of BC/(OC+sulfate) should increase and all else being equal, the plume would be more absorbing than reference conditions. In addition, if the reference conditions were causing a positive RF, then increasing aerosol levels would also likely cause a positive RF (all else being equal). I agree that in this case, increased OC overwhelms the BC signal (since RF results are negative) but I am not sure the explanation is as straightforward as this section makes it appear. As a result I don't think this part adds a lot to the understanding of the event and I suggest removing these lines, or making a stronger case for this.

38. P18 L 14: I suggest removing "We found" and adding references supporting these statements, since I don't think this was investigated in detail in the present study.

39. P18 L25: Markowicz et al. (2017) studied the same case, right? If they did, this should be mentioned explicitly here, e.g. "transport of this BB plume over the Northern Hemisphere" or "the same BB plume"

40. P18 L29: Can you say here why this comparison is only done over the ocean?

41. P18 L30; Can you remind why the input parameters for the models are different, i.e. MODTRAN uses LIDAR and radiosondes, Fu-Liou uses NAAPS etc.

42. P20 L13: "of MODTRAN and simulations is", "Fu-Liou" is missing from the text here?

43. P21 L 11: "3D distribution", maybe use instead "3D effects on radiative forcing" or something similar, since you don't show directly here "3D distributions" but 3D effects on RF.

44. P21 L 12: You can remind here where this single-cell is located.

45. P21 L 16-23: This section is confusing, consider giving a name to the simulations (e.g. Control and PP) and saying clearly at the beginning that you perform 2 simulations, one with and one without 3D effects. The way this section is worded makes it seems like "ICA" and "plane-parallel" are equivalent terms, but it is not clear to areader unfamiliar with these terms if this is true since

these terms are not clearly defined. If this is indeed the case, I suggest dropping most mentions of ICA later in the text, and only include "ICA" here when describing the "PP" simulation.

46. P21 L.25: RFcell_rel is not properly defined here and it is not explained why this quantity is needed in addition to RF_rel.

47. P22 L1: "the noise of the Monte-Carlo method may enhance it". Can you elaborate why?

48. P25 L16: Can you add a concluding remark discussing the interest or significance of these ILES results?

49. P25 L 27: This should be LIDAR instead of lidar.

---

## Author Response (AR2)

**Authors' Response to the Referee reports**

June 5, 2018

*Italic font style denotes the Referee comments,* while normal font - our answer.

The revised manuscript with tracking contains only the changes explicitly mentioned by the reviewers. All minor English corrections were skipped for the clarity.

**The Referee #1**

*P10, L24: magnitude of 10 —–> is this true? I can only read about 1 from the Figure 1.*

Thank you, we meant "the factor of 10" and we have changed the text accordingly.

*P10, L28: lidar data Markowicz —–> lidar data by Markowicz*

Agreed, changed as suggested.

*P11, Fig. 1: Label of ordinate is difficult to understand, t/a —–> t or a w/CL —–> w or CL*

The referee is right, we have changed the labels to alpha & tau etc.

*P12, L19-21: The sentence is slightly difficult to follow.*

The sentence was rephrased, as follows:

The advection of such humid air-masses may significantly enhance the water uptake of aerosols, hence their scattering properties. Using in situ instruments, that dry the particles (RH usually of around 15 % in the chamber), possibly leads to an appreciable underestimation of aerosol scattering, and thus radiative properties.

*P17. Fig. 4 and texts: Discrepancies between RF of surface radiometer observations and MODTRAN simulation are not clear yet.*

The following more detailed elaboration over the differences between radiometer and Modtran data was added to the section:

$RFs_{surf}$ were estimated by means of two approaches: we used MODTRAN (Mod $RF_{surf}$) simulations to account for both terms (representing polluted and clean cases respectively; for details see section **??**) in the following equation:

$$RF_{surf} = (F_{in} - F_{out}) - (F_{cin} - F_{cout}) \tag{1}$$

where $F_{cout}$ is total outgoing flux at the surface, simulated in the clean case. In the second approach, the radiometer data were used in place of the polluted case simulated by MODTRAN RTM. Since the second term of eq. 1 is identical in both $RFs_{surf}$ approaches, the mean discrepancies between Mod and Rad $RFs_{surf}$, exceeding 30 % during the event, relate to differences in Mod and Rad $F_{in}$ (in particular $F_{diff}$). Further to this, the 3D effects of the surface, uncertainty of the radiometers

enhanced by high solar zenith angles, and approximations used for the model of aerosol optical properties in the RTMs may play a major role.

**The Referee #2**

*General comment: This study investigating the intense radiative impacts in the Arctic (Ny-Ålesund, Svalbard) of an extreme biomass burning event is scientifically significant and of good scientific quality, and is in my opinion appropriate for ACP. However, I think there are serious issues with the English language. Specifically, I found that, while some sections (e.g. 2.2, 2.3, 3.7) are well written, most sections of the manuscript (and especially the introduction and Sections 3.4) are very difficult to understand even by someone already quite familiar with the topic. As a result, I think the paper fails to pass these ACP criteria 10- Is the overall presentation well structured and clear? 11- Is the language fluent and precise? I think the authors should have another look at the English language in the whole paper, and in the indicated sections in particular. Comments (line numbers and pages refer to the revised version of the manuscript with tracked changes)*

We wanted to thank the reviewer 2 for his/hers relevant comments. We were concerned about the readability of the paper, that is why it went through two separate English corrections by natives, taking into account both the scientific and linguistic issues. We hope that it helped to improve the paper.

**Specific comment**

*1. Title: "impact ....on the radiative forcing in the Arctic", the wording is strange, in my opinion it should be instead "Radiative forcing of a strong BB event" or "Radiative impact of a strong BB event"*

Thank you for this comment, the title was changed to: "Radiative Impact of an Extreme Arctic Biomass-Burning Event".

*2. Abstract: "the factor of 10 in comparison to the average" should be "a factor of 10 above the average"*

We have changed the text accordingly.

*3. P1 l17: "2.0 Pg of carbon aerosols is released into the atmosphere by fires each year", for now the text makes it seems like this includes all sources.*

Indeed, we have clarified the sentence by adding: [...] each year due to wildfires.

*4. P2 L3: "90 % of which are present in the fine mode (sizes XX nm - XX nm)"*

Thank you, however we feel this information is too detailed in this sentence.

*5. P2 L5-18: This section is unfortunately extremely difficult to read, please rephrase. 6. P2 L16: Please rephrase, BB plumes are darker than bright clouds or snow but do not directly decrease the albedo of clouds and snow (unless this sentence is about absorbing aerosols deposited on snow or cloud/aerosol interactions).*

In the revised manuscript the section is written as follows:

The presence of BB aerosol causes heating of the air layer in which the transport takes place. Regarding the columnar properties however, smoke existence results in a weak cooling at the top of the atmosphere (TOA) due to predominant scattering properties of the plume (Hansen et al., 2004). The magnitude of its impact on the radiative properties is nevertheless strongly dependent on the chemical composition of the smoke plume, due to the adversative radiative responses of the atmosphere exposed to black and organic carbon respectively, being negative for the latter (Myhre et al., 2013a).

A number of papers analysed the associated annual mean value of instantaneous clear-sky aerosol direct radiative forcing (RF) at the TOA ($RF_{toa}$) associated with BB plumes. Myhre et al. (2013a) presented the results from AeroComII's 28 models, indicating a global mean BB RFtoa on the level of of approximately -0.01 $\pm$ 0.08 Wm$-2$ . A similar value of 0.0 $\pm$ 0.2 Wm$-2$ was presented by Myhre et al. (2013b) in the Fifth Assessment IPCC Report. Despite a rather low (and negative) mean global value of BB RFtoa , on a regional scale (especially over bright surfaces), smoke may well play a substantial role in affecting radiative properties of the atmosphere (Wang et al., 2006). In the case of high surface albedo, the existence of smoke particles leads to albedo changes of the underlying clouds and the surface; and that in turn may the decrease of columnar albedo at the TOA. This may in turn indicate a positive RFtoa (Screen and Simmonds, 2010), leading to positive feedback within the entire atmospheric column. Based on AeroCom Phase II multi-model evaluations, Sand et al. (2017) found the annual median value of ensemble RFtoa in the Arctic region to be 0.01 Wm$-2$. Similar results are presented in Wang et al. (2014), who estimated its value at around 0.004 Wm$-2$ .

*7. P2 L24 This is also very unclear. "it is unlikely", what is "It" ?*

Thank you, we rephrased this sentence:

The significantly high RF uncertainty is associated mainly with the approximations of surface properties dependent on the daily and seasonal cycles, as well as the aerosol optical and microphysical properties which undergo ageing processes, whilst being transported across a large region (Bond et al., 2013; Ortiz-Amezcua et al., 2017; Koch et al., 2009; Janicka et al., 2017).

*8. P2 L28 Please define AERONET and add a reference*

Changed as suggested.

*9. P3 L26, I think you should use "in the vicinity of Ny-Ålesund (Svalbard)" here instead of "Kongsfjorden", since "Kongsfjorden", will not be known by most readers. I understand that this was implemented to remove "Ny-Ålesund valley" from the manuscript, but in most cases it seems like using "Ny-Ålesund" or "near Ny-Ålesund" would be clearer than using "Kongsfjorden".*

Changed as suggested.

*10. P3 L 27 A section number is missing.*

Inserting subsection title without a number was our intention.

*11. P4 L9-10: If simulation names are introduced, they should be mentioned later in the text when appropriate, e.g. when showing simulation results in Figure 3, mention that this is the 'polluted' simulation.*

Changed as suggested.

*12. P4 L7: I think you should mention there that thermodynamical variables are "from radiosonde measurements (see section ??)", for now it seems like you say that they are from HITRAN.*

Thank you, we have changed the text accordingly.

*13. P4 L 11: Say why you also use Fu-Liou, if MODTRAN is the main RTM used in the study.*

In the revised manuscript we added additional information about the purpose for presenting Fu-Liou results.

*14. P5 L15: This should also say what GEM-AQ is used for.*

Thank you, we have changed the text accordingly.

*15. P7 L22: What do you mean by "generalized"? Do you mean "regridded" or "upscaled"?*

We meant regridded and the text was changed as suggested.

*16. P9 L7: Does this apply to aged BB aerosols?*

Yes, see Lynch et al., 2016.

*17. P10 L 23: For clarity I suggest "were changed by a factor of 10"*

Thank you.

*18. P11 fig1: Replace "tau/alpha" by "tau & alpha", for now it seems like the ratio of the 2 is plotted. Same for omega/CL.*

Thank you, we have changed the text accordingly.

*19. P11 L6 "variability of alpha was rather stable" should be replaced by "variability of alpha was rather low" or "rather limited".*

Changed as suggested.

*20. P12 L26 remove "may" in "may support the above statement"*

Thank you, it was removed.

*21. P12 L30: Please rephrase, "exemplary" here is confusing, since this word usually means "perfect, exceptional" and also less commonly "providing an example".*

Thank you, we used 'example' instead.

*22. P13 Fig2. There is a problem with the legend and labels in this figure. For example the green line is supposed to be $\sigma_{ext}^a$ but is in fact only $\sigma_{ext}^a$ observed by LIDAR. The legend should say specifically that green is observed $\sigma_{ext}$, or lines in the second column (GEM-AQ) should also be in green. Columns should also be labeled to make this clearer, e.g. column 2 should be "GEM-AQ $\sigma_{ext}^a$", column 1 should be "observed ... "*

The GEM-AQ $\sigma_{ext}$ profile was changed to green and the figure added abbreviations of lidar and GEM-AQ data.

*23. P14 L1. It says there that in the last case vertical mixing is suppressed, when the following paragraph only mentions "the existence of vertical mixing". Where is this suppression shown or discussed?*

We mentioned about low-level cloud appearance in the subsection 3.1. A detailed discussion over the topic was previously published in Markowicz et al., 2016a, so we do not want to elaborate over it in this paper.

*24. P14 l6, replace "the next day" by "the 11th", since just above the "10th and 11th" are mentioned, which would mean that "the next day" is the 12th.*

Thank you, we have changed the text accordingly.

*25. P14 L17 Do you really mean "resulting from" (smoke mixing creates cumulus) or do you mean "resulting in" (cumulus lead to smoke mixing)?*

We meant "resulting in", thank you.

*26. P14 L20: Please rephrase the last sentence, which I could not understand as is.*

We rephrased the sentence to: $RFs_{surf}$ were estimated by means of two approaches: we used

MODTRAN (Mod $RF_{surf}$) simulations to account for both terms (representing polluted and clean cases respectively; for details see section **??**) in the following equation:

$$RF_{surf} = (F_{in} - F_{out}) - (F_{cin} - F_{cout}) \tag{2}$$

where $F_{cout}$ is total outgoing flux at the surface, simulated in the clean case. In the second approach, the radiometer data were used in place of the polluted case simulated by MODTRAN RTM. Since the second term of eq. 2 in both $RFs_{surf}$ approaches are identical, the mean discrepancies between Mod and Rad $RF_{surf}$, exceeding 30 % during the event, relate to differences in Mod and Rad $F_{in}$ (in particular $F_{diff}$). Further to this, the 3D effects of the surface, uncertainty of the radiometers enhanced by high solar zenith angles, and approximations used for the model of aerosol optical properties in the RTMs may play a major role.

*27. Section 3.3: I think you should mention here that (if I'm not mistaken) part of this good correlation is due to the daily cycle in insolation.*

This information is given later on, in the section 3.4, when we discuss the differences between radiometer and MODTRAN results.

*28. P15 L 2 I think the following would be clearer: "occasionally represent all-sky conditions"*

Thank you, we have changed the text accordingly.

*29. P15 L11 and elsewhere: Is "translation" the right term here? Maybe "correction"? This is also present later in the text.*

Changed as suggested.

*30. P15 L13-15 There's a problem here with how the parentheses are placed, for now it seems like radiometer measurements are labeled as "polluted simulation".*

The text was rephrased, see comment 26.

*31. P15 L20-24: this is also very confusing, consider separating into 2 sentences, one comparing model and measurements, and one comparing the 2 model simulations.*

The text was rephrased, as following:

Both measured by radiometer (hereinafter referred to as Rad) and modelled by MODTRAN (hereinafter referred to as Mod) $F_{in}$ are in rather good agreement, deviating on average by only $9.7 \ Wm^{-2}$ (2 %) from each other. The existence of aerosol indicates the mean decrease of $F_{in}$ by 0.4 % (Rad $F_{in}$), as well as 2.3 % (Mod $F_{in}$), as compared to the mean value of $F_{cin}$ (7:00 to 14:00 UTC on $9^{th}$ July). Measured and modelled $F_{out}$ indicate a very good agreement with a difference of less than 1 %, reaching on average $69.8 \ Wm^{-2}$ (Rad) and $69.4 \ Wm^{-2}$ (Mod).

*32. Section 3.4. The naming of the modeled quantities is confusing, since the simulations are called "polluted" and "reference", but the model results use the index "c" (e.g. $F_{in}^c$) for the "reference" simulation and no index for the polluted case. Maybe use $F_{in}^{pol}$ and $F_{in}^{ref}$ for clarity (or p and r). I suppose "c" stands for "clean" but you present this simulation as a "reference", not "clean" case. This is also not clear in the caption of Figure 4, where you say "no aerosol load" instead of "reference aerosols".*

We have changed the naming in the figure.

*33. P16 L13-19. This section is extremely hard to understand, please rephrase and correct the English.*

The text was rephrased, as following:

The highest decrease in Mod $F_{in}$ is visible on $10^{th}$ July as indicated by the observed maximum of $\tau_{550}$ during the BB event. The reduction of Mod $F_{in}$ exceeded 27 % for the summer background conditions (compare 7:00 - 14:00 UTC on $9^{th}$ and $10^{th}$ July). Additionally, higher temporal variability of Rad $F_{in}$ at the time, with respect to the previous day, is observed. It is likely to result from both a possible BB aerosol activation and increased turbulence. Further to this, a number of high- and mid-level cumulus clouds are reported around noon and in the afternoon (Markowicz et al., 2016a), which support the above statement.

*34. P 16 L 33-35. You mention 3 variables but then mention discrepancies between "both variables". What variables are these? Do you mean "all variables"?*

Yes, we meant all variables. The text was changed accordingly.

*35. P17 L2: There should be other / more general references (e.g. review papers, multimodel analysis, IPCC) than Stone et al. (2008) mentioning this, since this is a well-known feature of the RF of aerosols.*

Thank you, we added the reference.

*36. P17 L 10 "the RF sign at TOA"*

Changed as suggested.

*37. P17 L11-16: If I'm not mistaken, if both BC and OC increase a lot and sulfate increases less, than the ratio of BC/(OC+sulfate) should increase and all else being equal, the plume would be more absorbing than reference conditions. In addition, if the reference conditions were causing a positive RF, then increasing aerosol levels would also likely cause a positive RF (all else being equal). I agree that in this case, increased OC overwhelms the BC signal (since RF results are negative) but I am not sure the explanation is as straightforward as this section makes it appear. As a result I don't think this part adds a lot to the understanding of the event and I suggest removing these lines, or making a stronger case for this.*

We kindly disagree with the reviewer. We feel that these sentences are of significant relevance and they provide a good explanation to the results presented in the subsection. If the conclusions are not explicit, detailed descriptions might be found in Moroni et al., 2017.

*38. P18 L 14: I suggest removing "We found" and adding references supporting these statements, since I don't think this was investigated in detail in the present study.*

Changed as suggested.

*39. P18 L25: Markowicz et al. (2017) studied the same case, right? If they did, this should be mentioned explicitly here, e.g. "transport of this BB plume over the Northern Hemisphere" or "the same BB plume"*

Changed as suggested.

*40. P18 L29: Can you say here why this comparison is only done over the ocean?*

The sentence was rephrased as following:

In the following subsection, all $RFs$ were retrieved over the ocean area, near Ny-Ålesund (78.5 $^o$N, 9.5$^o$E), assuming a spectral surface albedo of the Fresnel reflection over a water body to eliminate discrepancies in the surface properties from our investigation.

*41. P18 L30; Can you remind why the input parameters for the models are different, i.e. MODTRAN uses LIDAR and radiosondes, Fu-Liou uses NAAPS etc.*

We skipped this information, as it was previously mentioned in the section 2.1.

*42. P20 L13: "of MODTRAN and simulations is", "Fu-Liou" is missing from the text here?*

The model name was placed in the next line. In the revised manuscript (version without tracking) it is correct.

*43. P21 L 11: "3D distribution", maybe use instead "3D effects on radiative forcing" or something similar, since you don't show directly here "3D distributions" but 3D effects on RF.*

Changed as suggested.

*44. P21 L 12: You can remind here where this single-cell is located.*

Changed as suggested.

*45. P21 L 16-23: This section is confusing, consider giving a name to the simulations (e.g. Control and PP) and saying clearly at the beginning that you perform 2 simulations, one with and one without 3D effects. The way this section is worded makes it seems like "ICA" and "plane-parallel" are equivalent terms, but it is not clear to a reader unfamiliar with these terms if this is true since these terms are not clearly defined. If this is indeed the case, I suggest dropping most mentions of ICA later in the text, and only include "ICA" here when describing the "PP" simulation.*

We have rephrased the section and introduced new names to the simulations.

*46. P21 L.25: $RF_{rel}$ is not properly defined here and it is not explained why this quantity is needed in addition to $RF_{rel}$.*

New sentences were added to the subsection 2.3.4, explaining the above.

$RF_{rel}$ and $RF_{rel}^{cell}$ have slightly different meanings. $RF_{rel}$ represents aerosol impact on the flux of solar energy absorbed by a unit area of an actual sloped surface. This quantity is of local relevance, i.e. to vegetation or changes in the surface temperature. $RF_{rel}^{cell}$ is relevant to the radiative budget of the whole atmospheric column. Moreover, it can be used to compare results from RTMs with different geometries.

*47. P22 L1: "the noise of the Monte-Carlo method may enhance it". Can you elaborate why?*

The following explanation was added:

The actual value of $RF$ variability over the sea may be even lower, because the noise of the Monte Carlo method may enhance it. Being a probabilistic technique where photons are traced on their random paths through the atmosphere, Monte Carlo is associated with random noise.

*48. P25 L16: Can you add a concluding remark discussing the interest or significance of these ILES results?*

The following concluding remarks were added:

The obtained ILES results help us understand the potential effects of a BB plume on atmospheric dynamics on a local scale. Furthermore, the observed local production of turbulence and the associated vertical motion, may in turn affect factors such as cloud cover and the coupling between the surface layer and the plume layer, with potential effects on larger-scale dynamics. Further simulations, including water vapor and cloud condensate, are needed to study such effects in more detail.

*49. P25 L 27: This should be LIDAR instead of lidar.*

Changed as suggested.

[revised manuscript text omitted]